# Pantothenate kinase 2 interacts with PINK1 to regulate mitochondrial quality control via acetyl-CoA metabolism

Yunpeng Huang [1,10,13], Zhihui Wan[1,11,13], Yinglu Tang [2,13], Junxuan Xu[1], Bretton Laboret[2], Sree Nallamothu[2], Chenyu Yang[3], Boxiang Liu [4], Rongze Olivia Lu[5,12], Bingwei Lu [6], Juan Feng[7], Jing Cao[3], Susan Hayflick [8], Zhihao Wu [2✉] & Bing Zhou [1,9✉]

Human neurodegenerative disorders often exhibit similar pathologies, suggesting a shared aetiology. Key pathological features of Parkinson's disease (PD) are also observed in other neurodegenerative diseases. Pantothenate Kinase-Associated Neurodegeneration (PKAN) is caused by mutations in the human PANK2 gene, which catalyzes the initial step of de novo CoA synthesis. Here, we show that *fumble* (*fbl*), the human *PANK2* homolog in *Drosophila*, interacts with PINK1 genetically. *fbl* and *PINK1* mutants display similar mitochondrial abnormalities, and overexpression of mitochondrial Fbl rescues PINK1 loss-of-function (LOF) defects. Dietary vitamin B5 derivatives effectively rescue CoA/acetyl-CoA levels and mitochondrial function, reversing the PINK1 deficiency phenotype. Mechanistically, Fbl regulates Ref(2)P (p62/SQSTM1 homolog) by acetylation to promote mitophagy, whereas PINK1 regulates *fbl* translation by anchoring mRNA molecules to the outer mitochondrial membrane. In conclusion, Fbl (or PANK2) acts downstream of PINK1, regulating CoA/acetyl-CoA metabolism to promote mitophagy, uncovering a potential therapeutic intervention strategy in PD treatment.

[1] State Key Laboratory of Membrane Biology, School of Life Sciences, Tsinghua University, Beijing 100084, China. [2] Department of Biological Sciences, Dedman College of Humanities and Sciences, Southern Methodist University, Dallas, TX 75275, USA. [3] Department of Statistical Science, Dedman College of Humanities and Sciences, Southern Methodist University, Dallas, TX 75275, USA. [4] Department of Genetics, Stanford University School of Medicine, Stanford, CA 94305, USA. [5] Department of Neurosurgery, Dell Medical School, University of Texas Austin, Austin, TX 78712, USA. [6] Department of Pathology, School of Medicine, Stanford University, Stanford, CA 94305, USA. [7] School of Pharmaceutical Sciences, Tsinghua University, Beijing 100084, China. [8] Department of Molecular & Medical Genetics, Oregon Health and Science University, Portland, OR 97201, USA. [9] Shenzhen Institute of Synthetic Biology, Shenzhen Institutes of Advanced Technology, Chinese Academy of Sciences, Shenzhen 518055, China. [10]Present address: Key Laboratory of Systems Health Science of Zhejiang Province, Hangzhou Institute for Advanced Study, University of Chinese Academy of Sciences, Hangzhou 310024, China. [11]Present address: Department of Laboratory Medicine, Beijing Obstetrics and Gynecology Hospital, Capital Medical University, Beijing Maternal and Child Health Care Hospital, Beijing 100026, China. [12]Present address: Department of Neurological Surgery, Brain Tumor Center, University of California San Francisco, California, CA 94143, USA. [13]These authors contributed equally: Yunpeng Huang, Zhihui Wan, Yinglu Tang. ✉email: zhihaowu@smu.edu; bing.zhou@siat.ac.cn

Degenerative nerve disease, or often called neurodegenerative disease, is an umbrella term for a spectrum of disorders that primarily affect the neurons in different regions of human central nervous system (CNS)[1]. Human neurodegenerations usually develop two major symptoms during disease progression, dementia, and movement disorder. While both features often appear in multiple human neurodegenerative diseases, one symptom is usually more dominant[2]. The commonalities in disease symptoms indicate a potential connection in at least some of their etiologies. For example, Parkinson's disease (PD) is the second most prevalent progressive neurodegenerative disease characterized by a set of symptoms including tremor, bradykinesia, rigid muscles, impaired posture and balance, loss of automatic movements, and alterations in writing/speech and sleep[3]. However, Parkinsonism is not unique in Parkinson's disease and can be found in other rare neurodegenerative diseases such as the Pantothenate Kinase-Associated Neurodegeneration (PKAN)[4]. Unfortunately, little effort has been conducted to understand the underlying mechanistic connection so far.

People have been striving for decades to clarify the pathogenesis of PD. Postmortem examinations indicated PD as a movement disorder caused by selective degeneration of dopamine (DA) neurons in the substantia nigra pars compacta[3]. Up to now PD treatment strategy remains limited and few biomarkers exist for PD diagnosis[5–8]. While about 10% of patients carry genetic mutations (single gene mutations), over 90% of patients are considered as sporadic, that is, with no known genetic variants or family history[9]. Thus, both genetic and environmental risk factors are proposed to participate in PD pathogenesis, and mechanisms behind them are intricate and largely mysterious[10–12].

In spite of these, overwhelming evidence has been accumulated and indicates a central role of mitochondrial dysfunction in PD pathophysiology[13]. The first observation leading to this connection was made from people exposed to MPTP (1-methyl-4-phenyl-1,2,3,6-tetrahydropyridine), a mitochondrial toxin, which can be metabolized to the mitochondrial complex I (C-I) inhibitor MPP + and causes acute-onset Parkinsonism[14]. Other supporting evidences include the continually and consistently identified respiratory complex dysfunction in PD animal models[15,16] and PD patient postmortem studies[17,18], as well as the observation that rotenone, a complex I inhibitor, is strongly associated with PD risk in epidemiological studies[19].

Under normal circumstances, dysfunctional mitochondria are usually monitored and eliminated by a surveillance system termed Mitochondrial Quality Control (MQC)[20]. The MQC collectively includes all levels of recovery mechanisms: stimulation of local synthesis and import of nucleus-encoded mitochondrial proteins[21]; AAA protease for aberrant protein degradation in both mitochondrial matrix and intermembrane space[22]; ubiquitination/proteasome machinery for the outer mitochondrial membrane (OMM) protein removal; mitochondrial derived vesicles (MDVs) for clearing localized damages[23]; fission/fusion regulation[24]; and mitophagy to eliminate the entire dysfunctional mitochondrion[25].

Among the heterogenous PD-associated genes that have been identified from the familial PD research, two candidate genes *PINK1* (encoding PINK1, PTEN-induced serine/threonine kinase 1) and *PRKN* (encoding Parkin, E3 ubiquitin ligase), associated with early onset PD[26,27]; and proteins their encode are characterized to play an essential role in maintaining mitochondrial function, monitoring mitochondrial damage and initiating mitophagy[10,11,28–30]. PINK1[26] and Parkin[31], were found acting in a linear pathway in genetic epistasis analyses, first from *Drosophila*[28,29,32] and later from mammalian cells[33]. PINK1 positions upstream of Parkin in regulating the MQC pathway. Biochemical studies have revealed that PINK1 can be stabilized on the outer membrane of damaged mitochondria, and this activates Parkin by directly phosphorylating both ubiquitin and Parkin; activated Parkin enacts ubiquitination on multiple targeted mitochondrial proteins and provides the poly-ubiquitination signals for recruiting autophagy receptors[34]. In addition, under physiological conditions or mild mitochondrial damage, the PINK1/Parkin pathway also directs the mitochondria-localized translation of certain nucleus-encoded respiratory chain complex (nRCC) mRNAs to help maintaining the normal mitochondrial oxidative phosphorylation (OXPHOS)[35].

As one of cellular metabolic hubs, mitochondria also play essential roles in synthesizing and regulating critical metabolites, including Coenzyme A (CoA) and acetyl-Coenzyme A (acetyl-CoA). Not surprisingly, a previous study has reported that both CoA and acetyl-CoA levels are significantly dropped in *PINK1* mutants[36]. Similar extents of CoA and acetyl-CoA reduction were found in PKAN, formerly called Hallervorden-Spatz syndrome[4,37], which is defective in pantothenate kinase 2 (PANK2), an enzyme catalyzing the first step of CoA synthesis. In human, there are four PANKs, with only PANK2 being mitochondrial. In *Drosophila*, the homolog of human *PANK2* is *fumble* (*fbl*) (encoding Fbl, Fumble or dPANK), and inhibiting Fbl activity also causes severe neurological defects and reproduces some of the Parkinsonism symptoms[38,39]. Conversely, D-pantethine, a homologue of pantothenate and an active moiety of CoA[15], can significantly benefit *fbl* LOF flies[39] and likely MPTP-induced PD rodents[15]. Thus, emerging evidence implicates a possible link between PD and PKAN, while the key mechanistic work establishing the regulation and function of CoA metabolism in PD and PD-associated genes in PKAN is still absent.

In this study, we uncovered a sophisticated and conserved interaction among PINK1, Parkin, and Fbl/PANK2, and their roles in the control of CoA/acetyl-CoA synthesis and mitophagy regulation. Fbl/PANK2 is subject to the mitochondria-localized translational regulation by PINK1 and Parkin, and Fbl/PANK2 overexpression can significantly suppress defects of *PINK1* LOF. This rescue is implemented by CoA and acetyl-CoA elevation and subsequently Ref(2)P (refractory to sigma P)/p62 (Sequestosome-1) acetylation. Our results help better understand the communality of the two diseases, PD and PKAN, and may facilitate the development of new treatment strategies in the future.

## Results

**Knocking-down *Drosophila fbl* results mitochondrial abnormalities resembling defects of *PINK1* LOF.** The *Drosophila fbl* mutant has been characterized to associate with multiple defects, including a significant drop of eclosion rate, abnormal wing morphology (wing inflation), progressive locomotor dysfunction, lifespan shortening, infertility, and broad neurodegeneration[38,40,41]. Among all *fbl* transcripts, Fbl-L (the longest isoform) encodes a mitochondria-targeted and sole active PANK. We thus suspected that Fbl might regulate mitochondrial functions, and consequently, we systematically analyzed possible mitochondrial defects under Fbl loss.

We first looked at the mitochondria morphology in dopaminergic (DA) neurons in *fbl* overexpression and knockdown adult flies. Prominent mitochondrial aggregations (labeled by mitoGFP) were observed in DA neurons of *fbl* RNAi flies, reminiscent of that in *PINK1* mutant (Fig. 1a, b), while *fbl-L* OE did not generate any notable abnormality (Fig. 1a, b). In addition, loss of DA neurons in the protocerebral posterior lateral (PPL1) cluster (in the posterior inferiorlateral protocerebrum near the posterior face of the brain) was seen in *fbl* RNAi flies, which also recaptured the *PINK1* mutant phenotype (Fig. 1c). Here, the

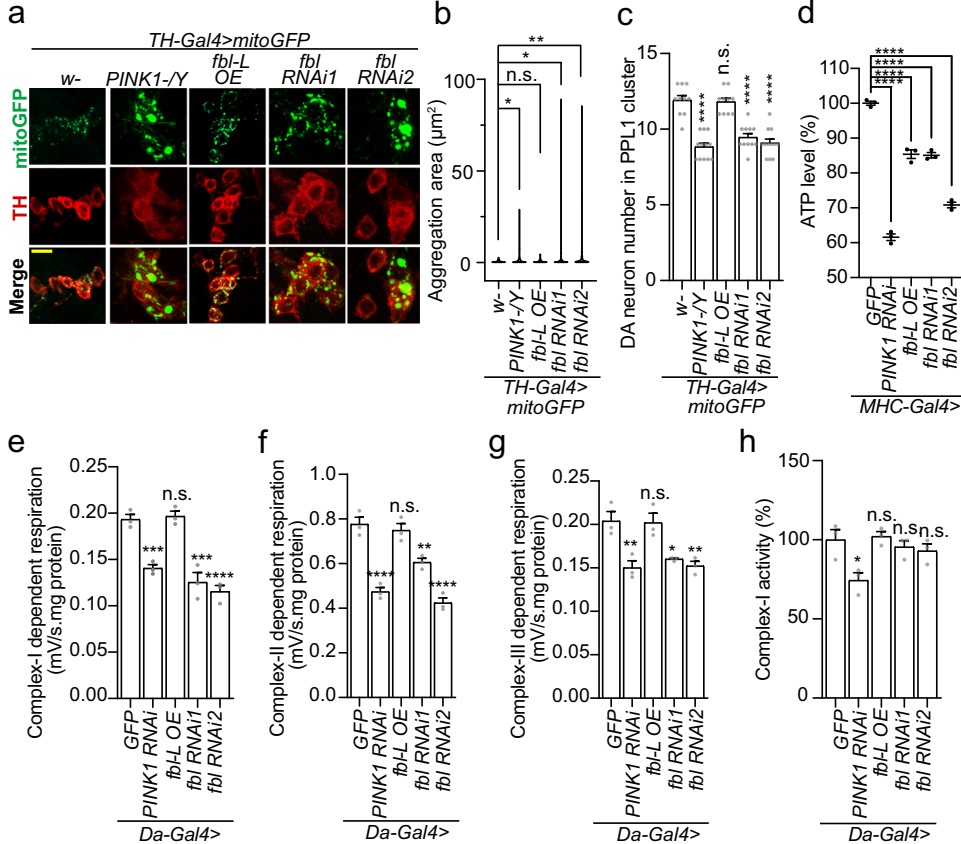

**Fig. 1 Fbl Regulates Mitochondrial Functions in *Drosophila*. a** Immunostaining showing effects of *fbl* on mitochondrial morphology in TH-positive dopaminergic (DA) neurons. WT (*w-/Y*) and *PINK1* mutant (*PINK1-/Y*) serves as the negative and positive controls. Scale bar, 10 μm. In this and all subsequent immunostaining figures of DA neurons, mitochondrial morphology is monitored by the mitoGFP reporter. **b** Violin plots showing the quantification of (**a**). In this and all subsequent mitochondrial quantifications in neurons, the significance was calculated by using two-proportion *Z* test, and the threshold was set to 3 μm². *n* = 3 biologically independent samples. **c** Effects of *fbl* on DA neuron loss in fly PPL 1 cluster. In this and all subsequent DA neurons counts. *n*=10 (*w-*, *fbl-L* OE) and 12 (other genotypes) biologically independent animals. **d** Effects of *fbl* on ATP level in fly indirect flight muscle. *n* = 3 biologically independent samples. **e–g** Effects of *fbl* on complex-I, -II and -III dependent respirations. **h** Effects of *fbl* on complex-I activity. For assays in (**d–h**), *n* = 3 biologically independent samples. For all assays in (**b–h**), *w-* or *UAS-GFP* was used as control, and the significance was calculated by using one way ANOVA followed by post hoc Dunnett's multiple comparation test. Data are presented as mean values ± SEM; n.s. not statistically significant; *$p < 0.05$; **$p < 0.01$; ***$p < 0.001$; ****$p < 0.0001$. Source data are provided as a Source Data file.

efficiencies of *fbl* OE and RNAi lines were confirmed by immunoblotting (Supplementary Fig. 1a, b) and RT-PCR (Supplementary Fig. 1c, d). Notably, neuronal tissue seemed more sensitive to Fbl deprivation, since mitochondria in the indirect flight muscle did not display any severe defect in *fbl* RNAi flies, unlike those in the *PINK1* mutant (Supplementary Fig. 1e–g). We next analyzed the mitochondrial physiology in *fbl* OE and RNAi flies. Thoracic ATP levels in *fbl-L* OE and *fbl* RNAi flies were both reduced, indicating the existence of an essential and sophisticated regulation of mitochondrial function by Fbl (Fig. 1d). To address the underlying basis of ATP reduction in *fbl* RNAi flies, we next purified the mitochondria from neuromuscular tissue of each genotype and measured their respiration abilities of individual complex by oxygen microelectrode. Moderate decrease of complex-I, -II, and -III dependent respiration was observed in *fbl* RNAi flies (Fig. 1e–g), also similar to that in the *PINK1* RNAi fly. Differing from *PINK1* RNAi, *fbl* RNAi did not obviously change the activity of complex-I (Fig. 1h), suggesting the existence of certain dissimilitude between the two. Altogether, *fbl* LOF recapitulates many mitochondrial defects previously found in the *PINK1* LOF model, prompting us to investigate the possible functional connection between *fbl* and *PINK1*.

**fbl genetically interacts with PINK1 in regulating mitochondrial homeostasis.** To investigate the genetic interaction between *fbl* and *PINK1* we tuned the *fbl* expression in *PINK1* LOF flies. We first found that OE of the mitochondrial isoform – Fbl-L, but not that of the cytosolic isoforms – Fbl-S1 or Fbl-S2, in indirect flight muscles using MHC-Gal4[29,42], dramatically rescued the abnormal wing posture (Fig. 2a) as well as the jumping/flight defect (Supplementary Fig. 2a) in *PINK1* RNAi flies. Conversely, *fbl* RNAi significantly aggravated the pre-existing *PINK1* RNAi wing abnormality (Fig. 2a and Supplementary Fig. 2a). While in wild type (WT) background, neither *fbl* short forms OE nor *fbl* RNAi would cause wing posture or movement defects (Supplementary Fig. 1f, g). Next, we measured the ATP production in fly muscles. Consistent with phenotypical changes, the ATP measurement also indicated that *fbl-L* OE partially restored the mitochondrial function in *PINK1* RNAi animals, whereas *fbl* RNAi exacerbated it (Fig. 2b). The effects of Fbl on tissue integrity (Fig. 2g) correlated well with effects on mitochondrial morphology, with *fbl-L* OE rescuing the mitochondrial aggregation phenotype in neuromuscular tissues caused by *PINK1* LOF, whereas *fbl-S* isoforms had no effect and *fbl* knockdown had the opposite effect (Fig. 2c–f and Supplementary Fig. 2b, c).

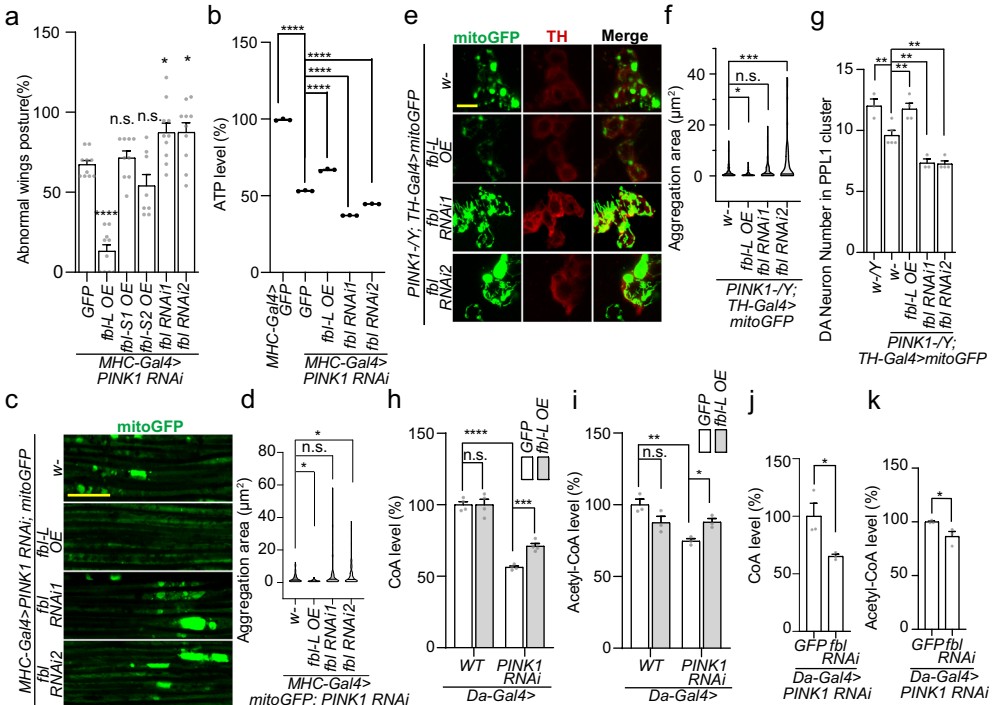

**Fig. 2 Genetic Interaction Between *fbl* and *PINK1*. a** Effects of Fbl variants on wing posture defect caused by *PINK1* RNAi in the indirect flight muscles. 25 flies per genotype per group were tested; $n = 8$ (*fbl-S2* OE), $n = 9$ (*fbl-S1* OE) and $n = 10$ (other genotypes) biologically independent groups. **b** Effects of Fbl on ATP level reduction caused by *PINK1* RNAi. In this and all subsequent ATP measurement figures, $n = 3$ biologically independent samples. For assays in (**a**, **b**) *UAS-GFP* serves as control, and the significance was calculated by using one way ANOVA followed by post hoc Dunnett's multiple comparison test. **c**, **e** Immunostainings showing effects of Fbl on mitochondrial aggregations in flight muscles (**c**) and DA neurons (**e**) caused by *PINK1* LOF. Scale bars: (**c**) 25 μm; (**e**) 10 μm. In this and all subsequent immunostaining figures of fly muscle and DA neurons, mitochondrial morphology is monitored with a mitoGFP reporter. **d**, **f** Violin plots showing the quantifications corresponding to (**c**) and (**e**). In mitochondrial quantifications in muscle and neurons, the significance was calculated by using two-proportion $Z$ test, the threshold was set at 3 $\mu m^2$, and $n = 3$ biologically independent samples. **g** Effects of Fbl on DA neurons' loss in PPL 1 cluster of *PINK1* mutant. $n = 3$ (*w, PINK1/fbl RNAi1*), $n = 4$ (*PINK1/ fbl* OE and *PINK1/fbl RNAi2*), and $n = 5$ (*PINK1*) biologically independent animals. **h**, **i** Effects of Fbl ectopic expression on CoA and acetyl-CoA levels in WT and *PINK1* RNAi flies. **j**, **k** Effects of *fbl* RNAi on CoA and acetyl-CoA levels in *PINK1* RNAi flies. $n=4$ biologically independent samples in assays (**h**) $n = 3$ biologically independent samples in assays (**j–k**). For assays in (**g**), the significance was calculated by using one way ANOVA followed by post hoc Šídák's multiple comparison test. For assays in (**h–k**) the significance was calculated by using two tailed unpaired t-test. Data are presented as mean values ± SEM; n.s., not statistically significant; *$p < 0.05$; **$p < 0.01$; ***$p < 0.001$; ****$p < 0.0001$. Source data are provided as a Source Data file.

The kinase activity of Fbl was necessary in the rescue, since a kinase-dead form of Fbl-L (Fbl-L K221A-2 and K221A-7) showed no rescue on wing posture abnormality (Supplementary Fig. 2d). In the *PINK1* knockdown but not WT flies, *fbl* OE increased the CoA and acetyl-CoA levels, indicating a global improvement of mitochondrial health in mutants (Fig. 2h, i)[36]. By contrast, *fbl* RNAi decreased the CoA and acetyl-CoA levels in WT flies (Supplementary Fig. 2f, g), and further in *PINK1* RNAi flies (Fig. 2j, k). Essentiality of CoA metabolism in the PINK1 related pathology was further confirmed by the observation that knocking down fly *Ppcdc*, a downstream enzyme of CoA synthesis pathway, could largely reverse the rescuing effect of *fbl-L* OE (Supplementary Fig. 2e). Consistent with genetic interactions, reduced levels of CoA and acetyl-CoA were detected in *fbl* RNAi, *PINK1* RNAi and *Ppcdc* RNAi larvae, while *Ppcdc* knockdown resulted in more severe reduction and pupal lethality (Supplementary Fig. 2f, g).

Previous studies indicated that *PINK1* LOF animals displayed impaired respiratory chain complex (RCC) assembly and mitochondrial respiration deficits[43,44]. In our assays, we found a restoration of the complex assembly of multiple RCCs – complex-II, -III, -IV, -V and super complexes by *fbl-L* OE in the *PINK1* mutant (Supplementary Fig. 2h), which was concordant with recovery of individual complex dependent respirations (Supplementary Fig. 2i–k). Worthy of mention, *fbl-*

*L* OE did not increase the activity of complex-I in the *PINK1* mutant (Supplementary Fig. 2l) or dramatically restore complex-I assembly (Supplementary Fig. 2h), which echoed well the previous observation that *fbl* knockdown did not affect the complex-I activity (Fig. 1h).

In contrast to the rescuing effect of Fbl on *PINK1* loss, *PINK1* OE failed to alleviate *fbl* LOF defects, e.g. wing inflation (Supplementary Fig. 2m) and survival (Supplementary Fig. 2n), suggesting Fbl acting downstream to PINK1. Hence, it appears that *fbl* epistatically interacts with *PINK1* to sustain mitochondrial health and maintain the tissue integrity.

**D-pantetheine rescues *PINK1* LOF defects.** Besides genetic manipulations, pharmacological treatments also lent strong support to the connection between PD pathogenesis and CoA synthesis. Previous studies revealed D-pantethine (D-pan), a close relative of vitamin B5 (pantothenate) and a metabolic substrate for CoA synthesis[39], could bypass PANK in CoA synthesis in a currently unclear biochemical pathway, and nicely rescue multiple animal PKAN models[39,45,46]. We administered D-pan through diet to WT and *PINK1* LOF flies. Similar to the genetic studies, D-pan treatment significantly rescued *PINK1* LOF phenotypes, including abnormal wing posture (Fig. 3a and Supplementary Fig. 3b), flight activity drop (Supplementary

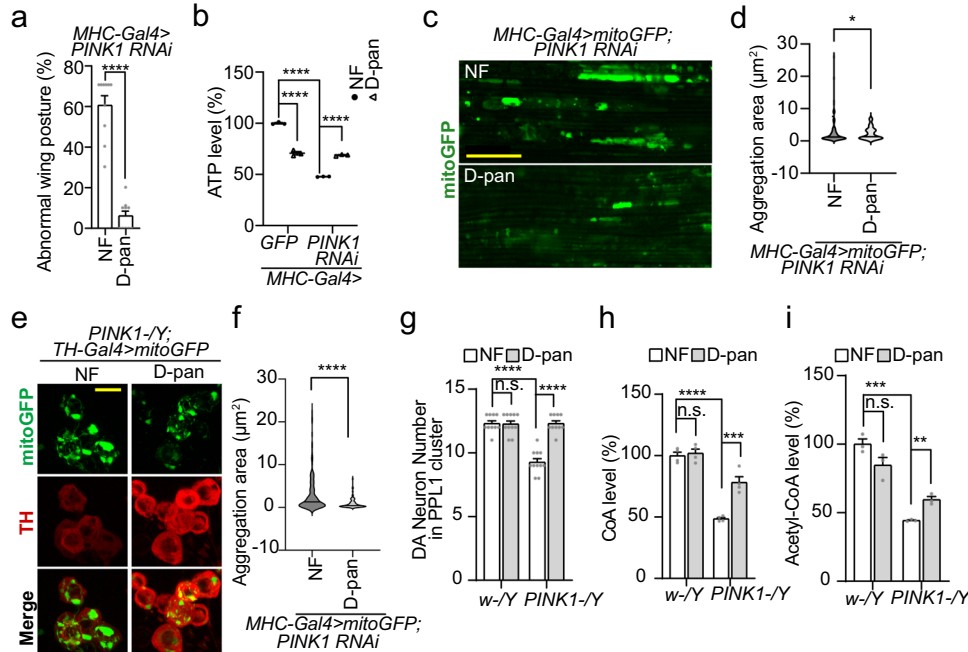

**Fig. 3 D-pantethine Rescues *PINK1* LOF Defects. a** Rescue of D-pantethine (D-pan) alimentation on wing posture defect in *PINK1* RNAi flies. 25 flies per genotype per group were tested, *n* = 10 biologically independent groups. In this and following figures in Fig. 3, NF, as normal food. **b** Effects of D-pan alimentation on ATP levels in WT and *PINK1* RNAi flies. *UAS-GFP* serves as control. *n* = 3 biologically independent samples. **c**, **e** Immunostaining showing effects of D-pan alimentation on mitochondrial aggregations in flight muscle (**c**) and DA neurons (**d**) caused by *PINK1* LOF. Scale bars: (**c**) 25 μm; (**d**) 10 μm. **d**, **f** Violin plots showing the quantifications corresponding to (**c**) and (**e**). The significance was calculated by using two-proportion *Z* test, and thresholds were set at 8 μm² (**d**) and 3 μm² (**e**); *n* = 3 biologically independent samples. **g** Rescue of D-pan alimentation on DA neurons loss in PPL 1 cluster of *PINK1* mutants. *n* = 9 (*w-*/NF, *PINK1*/D-pan), *n* = 10 (*w-*/D-pan, *PINK1*/NF) biologically independent animals. **h**, **i** Effects of D-pan alimentation on CoA and acetyl-CoA levels in WT (*w-*/Y) and *PINK1* mutant. *n* = 4 biologically independent samples in assay (**h**) *n* = 3 biologically independent samples in assay (**i**). For assays in (**a**, **b**, **g–i**), the significance was calculated by using two tailed unpaired *t*-test. Data are presented as mean values ± SEM; n.s. not statistically significant; *$p < 0.05$; **$p < 0.01$; ***$p < 0.001$; ****$p < 0.0001$. Source data are provided as a Source Data file.

Fig. 3c–e), thoracic ATP level reduction (Fig. 3b), mitochondrial aggregation (Fig. 3c–f) and DA neuron loss (Fig. 3g). We also observed elevated CoA and acetyl-CoA levels (Fig. 3h, i), which was consistent with the previous report[39] and correlated well with the improved mitochondrial respirations (Supplementary Fig. 3f-3h). Again, we did not find any evidence of promoting complex-I activity (Supplementary Fig. 3i) or assembly (Supplementary Fig. 3j) by D-pan alimentation. Furthermore, 4'-phospho-pantetheine (4'-PPT), known to be the direct precursor of CoA, but not CoA itself (probably due to its poor bioavailability), significantly rescued the abnormal wing posture and extended lifespan in *PINK1* mutant (Supplementary Fig. 3k, l). In addition, fosmetpantotenate (RE-024), another precursor of CoA, moderately but significantly rescued the abnormal wing posture and large mitochondrial aggregations in *PINK1* LOF flies (Supplementary Fig. 3m–o). Similar effects of D-pan (D-pantethine), 4'-PPT (4'-phosphopantetheine) and RE-024 (fosmetpantotenate) strongly indicate that the rescue observed is via the CoA synthesis pathway. Combining the genetic and pharmacological rescue data, we believe that the benefit of *fbl-L* OE and D-pantethine diet supplement in *PINK1* LOF come from their ability of enhancing the CoA synthesis pathway.

**Fbl's rescue is in parallel with parkin, downstream of PINK1.** Parkin has been reported to act downstream of PINK1 and play a critical role in MQC and maintenance of tissue integrity[28,47]. *parkin* mutant shows defects akin to the *PINK1* mutant[28,29,32,47]. We thus evaluated effects of *fbl* OE and D-pan in the *parkin* mutant. Surprisingly, neither Fbl-L OE nor D-pan alimentation

could significantly rescue the abnormal wing posture (Fig. 4a, b), mitochondrial aggregation (Fig. 4c–f) and depressed ATP production (Fig. 4g, h) in the *parkin* mutant. No increment of CoA or acetyl-CoA level was observed either after increasing *fbl* expression (Fig. 4i, j). Surprisingly, unlike in *PINK1* LOF (Fig. 2h, i), the *parkin* mutant exhibited an elevated CoA level but a declined acetyl-CoA level (Fig. 4i, j).

It has been reported that *PRKN* knockdown in human H460 cells impairs the activity of PDH (pyruvate dehydrogenase) complex[48], which could be the reason for CoA accumulation and acetyl-CoA decline in fly *parkin* mutant. In order to test this hypothesis, we knocked down (RNAi) and overexpressed *Pdha* (encoding Pdha, Pyruvate dehydrogenase E1 alpha subunit) in *parkin* mutant adult flies via the GeneSwitch GAL4 system (because *Pdha* knockdown severely influenced fly development), and compared their CoA and acetyl-CoA levels. *Pdha* knockdown strongly enhanced CoA accumulation, while *Pdha* overexpression only mildly mitigated it. Interestingly, when *parkin* was suppressed *Pdha* knockdown further reduced the acetyl-CoA level and *Pdha* overexpression rescued it (Supplementary Fig. 4a, b). As an important control, in the *parkin* mutant, the level of Pdha protein was not significantly reduced (Supplementary Fig. 4c).

These findings suggest a distinctiveness of these two genes, *PINK1* and *parkin*, in modulating CoA/acetyl-CoA synthesis and mitochondrial metabolic homeostasis, despite their tight connection in the MQC regulation. To further study their interaction in this regard, we measured the CoA and acetyl-CoA levels in different combinations of *PINK1* and *parkin*. Intriguingly, both *parkin* OE and RNAi notably restored the CoA reduction in *PINK1* RNAi flies, whereas had the opposite effect on the acetyl-

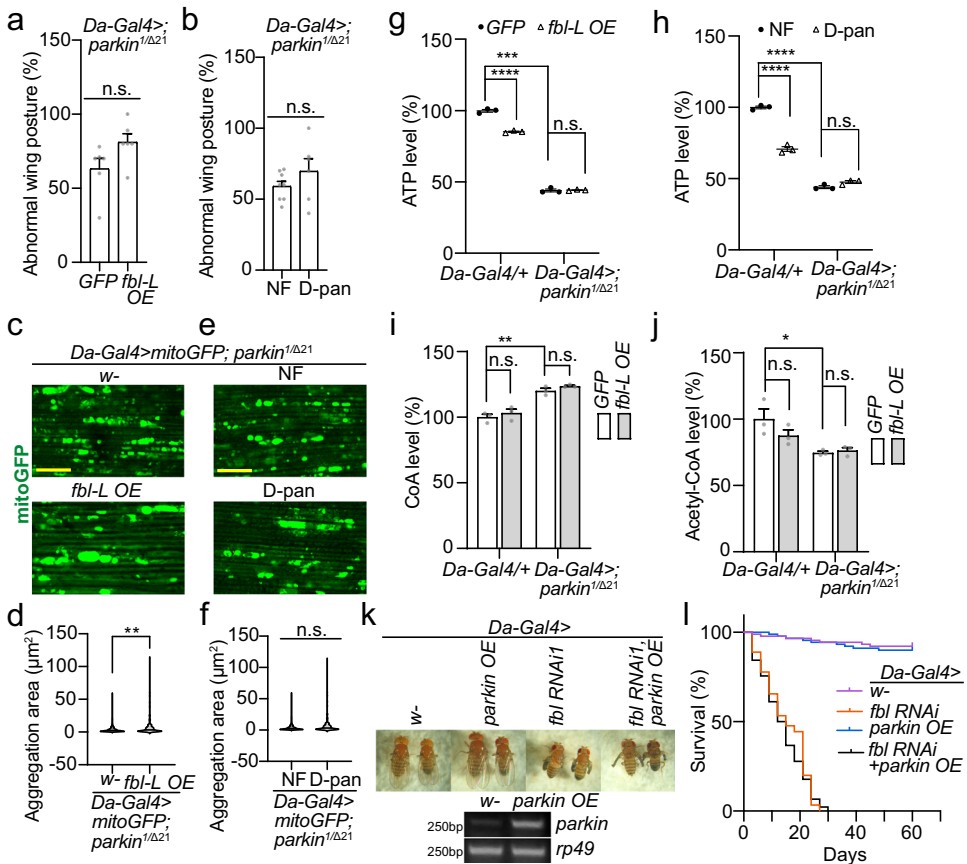

**Fig. 4 _fbl_ Does Not Genetically Interact with _parkin_. a, b** No rescue effects of _fbl-L_ OE (**a**) and D-pan alimentation (**b**) on wing posture defects in _parkin_ mutant. 25 flies per genotype per group were tested; _n_ = 6 biologically independent groups in assay (**a**); _n_ = 8 (NF), _n_ = 6 (D-pan) biologically independent groups in assay (**b**). **c, e** No rescue effects of _fbl-L_ OE (**c**) and D-pan alimentation (**e**) on mitochondrial aggregation defects in _parkin_ mutant muscles. Scale bars, 25 μm. **d, f** Violin plots showing the quantifications corresponding to (**c**) and (**e**). The significance was calculated by using two-proportion _Z_ test, and the threshold was set at 3 μm$^2$; _n_ = 3 biologically independent samples in assays (**d**) and (**f**). **g, h** No effects of _fbl-L_ OE (**g**) and D-pan alimentation (**h**) on ATP level reduction in _parkin_ mutant. **i, j** No effects of _fbl-L_ OE on CoA (**i**) and acetyl-CoA (**j**) levels in _parkin_ mutant. _n_ = 3 biologically independent samples in assay (**g–j**). For assays in (**a, g–j**), _UAS-GFP_ serves as control. **k** No rescue effect of _parkin_ OE on wing abnormal inflation caused by _fbl_ RNAi, and RT-PCR showing the efficiency of _parkin_ OE. _rp49_ serves as loading control. **l** Effects of _parkin_ OE on survival curve decline caused by _fbl_ RNAi. 30 flies per group and 3 independent groups (90 flies in total) were tested per genotype. For assays in (**a, b, g–j**), the significance was calculated by using two tailed unpaired t-test. For assays in (**l**) the significance was calculated by Log-rank (Mantel–Cox) test. Data are presented as mean values ± SEM; n.s. not statistically significant; *_p_ < 0.05; **_p_ < 0.01; ***_p_ < 0.001; ****_p_ < 0.0001. Source data are provided as a Source Data file.

CoA level (Supplementary Fig. 4d, e). In line with the elevated CoA level in the _parkin_ mutant, similar to D-pan, 4'-PPT could not rescue the abnormal wing posture and lifespan in _parkin_ mutant (Supplementary Fig. 4f, g). Lastly, _parkin_ OE did not ameliorate those defects caused by _fbl_ RNAi in our assays, such as wing inflation and shortened lifespan (Fig. 4k, l). Taken together, Parkin and Fbl act somewhat independently downstream of PINK1 in regulating CoA metabolism and maintaining the normal mitochondrial homeostasis.

**Fbl rescues mitochondrial damages in _PINK1_ mutant through mitophagy.** On damaged mitochondria, PINK1 phosphorylates Parkin and the activated Parkin subsequently generates poly-ubiquitination signal on some OMM (outer mitochondrial membrane) proteins in their first wave of actions[49]. Autophagy receptors can then bind with ubiquitinated targets and trigger following mitophagy processes. To investigate the role of Fbl and CoA metabolism in mitophagy induction, we examined their effects on p62, the sole autophagy receptor Ref(2)P in _Drosophila_[50], and autophagy-related proteins (Atg proteins)[51,52]. In the genetic analysis, we found that in _PINK1_ RNAi back-ground, knocking-down _ref(2)P_ or _Atg1_ (encoding Atg1,

Autophagy-related 1) could effectively diminish rescue effects from _fbl_ OE; while OE of _ref(2)P_ or _Atg1_ could significantly neutralize the severity enhanced by _fbl_ RNAi in multiple aspects: wing posture (Fig. 5a), ATP production (Fig. 5b), and mito-chondrial aggregation (Fig. 5c). Besides _Atg1_, genetic regulations of other _Atg_ genes were also able to occlude the beneficial effects of _fbl_ OE, indicating a nexus of Fbl with mitophagy (Supplementary Fig. 5a). To evaluate the promoting effect of Fbl on mitophagy, we examined the selected mitophagy marker Marf (Mitochondrial assembly regulatory factor, homolog of human Mitofusin-2)[53]. We found that Marf was effectively eliminated by _fbl_ OE in both WT and _PINK1_ mutant flies (Supplementary Fig. 5b). In line with it, our semiquantitative RT-PCR data of analyzing the transcription of mitochondrial genome encoded genes such as _mt:CoI_ (encoding mt:CoI, mitochondrial Cyto-chrome c oxidase subunit I), _mt:Cyt-b_ (encoding mt:Cyt-b, mitochondrial Cytochrome b) and _mt:ATPase6_ (encoding mt:ATPase6, mitochondrial ATPase subunit 6), showed that _fbl_ OE decreased the mitochondrial mass, while _fbl_ RNAi increased it (Supplementary Fig. 5c, d), without affecting the transcription of _srl_ (_spargel_, encoding spargel, homolog of human PGC-1α) and mitochondrial biogenesis (Supplementary Fig. 5g, h). In addition,

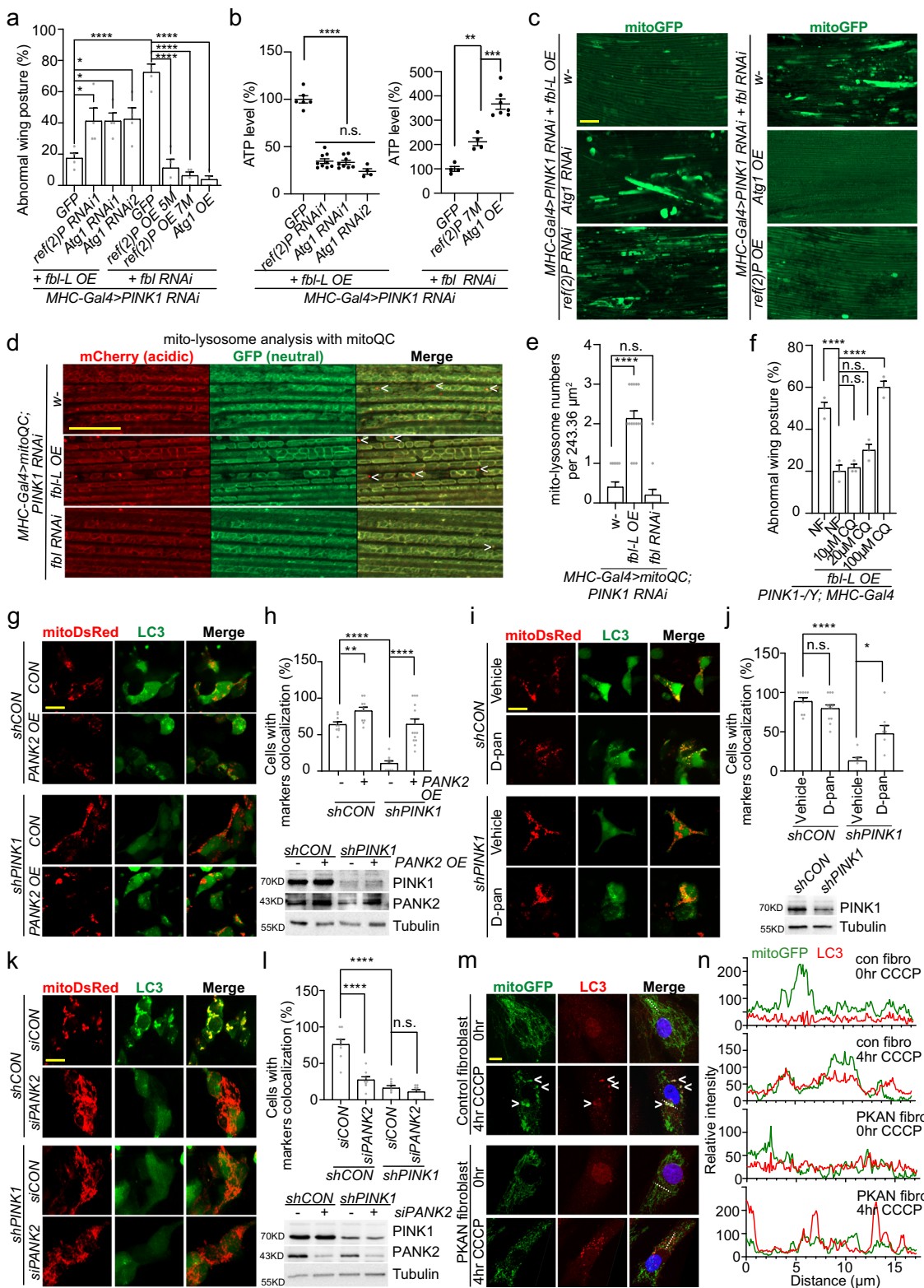

consistent with genetic interactions (Supplementary Fig. 5a, b), RT-PCR analyses also revealed similar changes in mitochondrial mass, that is, mitophagy stimulated by *fbl* OE could be impeded by *Atg1* and *ref(2)P* RNAi while enhanced by *Atg1* OE (Supplementary Fig. 5e, f), without altering the mitochondrial biogenesis (Supplementary Fig. 5i, j).

To further study the involvement of mitophagy in Fbl's rescue, we first visualized the occurrence of mitophagy (mitochondrial degradation in lysosome) in fly muscles by ectopically expressing the mitoQC indicator[54]. In *PINK1* RNAi flies with *fbl* OE, mito-lysosomes, indicated by the signal of "spectral shifted" puncta of more acidic conditions, were found dramatically increased (Fig. 5d, e). Consequently, more ATG8a (Autophagy-related 8a)/LC3 (Microtubule-associated proteins 1 A/1B light chain 3 beta) signals were found to colocalize with the mitochondrial marker – ATP5α (ATP synthase F1 subunit alpha) when *fbl* was

**Fig. 5 Fbl Helps Eliminate Damaged Mitochondria via Mitophagy. a** Effects of *ref(2)P* and *Atg1* on wing posture defect showing their genetic interactions with *fbl* in *PINK1* RNAi flies. 25 flies per genotype per group were tested, $n = 6$ (*fbl-L OE/GFP*), $n = 9$ (*fbl-L OE/ref2(P) RNAi1*), $n = 8$ (*fbl-L OE/Atg1 RNAi1*), $n = 4$ (*fbl-L OE/Atg1 RNAi2*), $n = 6$ (*fbl-L OE/GFP, fbl RNAi/GFP, fbl RNAi/ref2(P) 7 M, fbl RNAi/Atg1 OE*) biologically independent samples. **b** Effects of *ref(2)P* and *Atg1* on ATP level regulation with *fbl* in *PINK1* RNAi flies. For assays in (**a**, **b**) *UAS-GFP* serves as control. **c** Effects of *ref(2)P* and *Atg1* on mitochondrial aggregations in flight muscle with *fbl* in *PINK1* RNAi flies. Scale bar, 25 µm. *w-* serves as control. **d** Mito-lysosome analysis with mitoQC in *PINK1* RNAi flies showing the regulation of *fbl* on mitophagy. Scale bar, 25 µm. **e** Quantification of mito-lysosomes shown in **d** $n = 15$ from 5 biologically independent animals. *w-* serves as control. **f** Effect of chloroquine (CQ) on wing posture showing its inhibition of *fbl's* rescue in *PINK1* RNAi flies. 25 flies per genotype per group were tested, $n = 3$ biologically independent groups. **g**, **i**, **k** Immunostaining of mitoDsRed and LC3 showing effects of human *PANK2* OE, **g** D-Pan alimentation (**i**) and *PANK2* RNAi (**k**) on mitophagy induction in WT and *PINK1* knockdown (shRNA) HEK cells upon CCCP treatment. Scale bars, 10 µm. **h**, **j**, **l** Quantifications of **g**, **i**, **k** correspondingly. $n = 80/9$, means 80 cells examined over 9 independent experiments (*shCON/CON*), $n = 70/10$ (*shCON/PANK2 OE*), $n = 65/9$ (*shPINK1/CON*), $n = 80/14$ (*shPINK1/PANK2 OE*) in assay (**h**); $n = 60/9$ (*shCON/vehicle*), $n = 60/12$ (*shCON/D-pan*), $n = 41/7$ (*shPINK1/vehicle*), $n = 38/8$ (*shPINK1/D-Pan*), in assay (**i**); $n = 56/8$ (*shCON/siCON*), $n = 70/8$ (*shCON/siPANK2*), $n = 71/8$ (*shPINK1/siCON*), $n = 71/8$ (*shPINK1/siPANK*) in assay (**j**). Immunoblots embedded in (**h**, **j**, and **l**) showing gene knockdown and overexpression efficiencies in experiments (**j**, **i**, and **k**) accordingly. **m** Immunostaining of mitoGFP and LC3 showing the mitophagy decline caused by *PANK2* LOF in human control and PKAN fibroblasts upon CCCP treatments. Scale bar, 10 µm. Arrow heads, colocalization of mitoGFP and LC3. **n** Colocalization analysis of **m**. For assays in (**a**, **f**), the significance was calculated by using one way ANOVA followed by post hoc Šídák's multiple comparation test (**a**) or Dunnett's multiple comparisons test (**f**). For assays in (**b**, **e**, **h**, **j**, **l**), the significance was calculated by using two tailed unpaired *t*-test. Data are presented as mean values ± SEM; n.s. not statistically significant; *$p < 0.05$; **$p < 0.01$; ***$p < 0.001$; ****$p < 0.0001$. Source data are provided as a Source Data file.

overexpressed, and aloof from each other when *fbl* expression was suppressed (Supplementary Fig. 5k). To verify the biological significance, we fed flies with chloroquine (CQ), a classic autophagy inhibitor which impairs autophagosome fusion with lysosomes, resulting in the accumulation of undegraded cargoes in cells[55]. Notably, CQ administration in *PINK1* mutant flies sufficiently inhibited the wing posture rescue of *fbl* OE (Fig. 5f) and degradation of autophagy markers e.g. poly-ubiquitin and Ref(2)P (Supplementary Fig. 5l). Altogether, our data in *Drosophila* models strongly suggest that, in *PINK1* LOF flies, Fbl helps maintain the mitochondria integrity via increasing the activity of mitophagy. Intriguingly, although Fbl modulates mitophagy in our study, neither overactivation nor inhibition of autophagy factors (*Atg* genes) and autophagy receptor (*ref(2)P* gene) could rescue defects in *fbl* knockdown flies, indicating that compromised mitophagy or autophagy may not be the main cause of PKAN pathogenesis (Supplementary Fig. 5m).

**The mechanism of Fbl/PANK2 assisting mitophagy is conserved**. We next investigated whether the uncovered connection between *fbl/PANK2* and *PINK1* holds true in human cells. To verify the occurrence of mitophagy in human cells, multiple autophagy markers including LC3, p62 and Lamp2 (Lysosomal-associated membrane glycoprotein 2), were counterstained with mitochondrial indicators in both WT and *PINK1* knockdown cells after 12 hours carbonyl cyanide m-chlorophenyl hydrazone (CCCP) treatment. We found that *PANK2* OE did not or at most only mildly facilitated the colocalization between autophagy markers and mitochondria in WT cells whereas dramatically restored the action of mitophagy in *PINK1* knockdown cells (Fig. 5g, h and Supplementary Fig. 6a–d). Immunoblotting also validated prompt degradation of autophagy receptors (p62, Optineurin/OPTN) and mitochondrial markers (Tom20 and Tim23), and LC3 lipidation (LC3-I/LC-II), indicating burst of mitophagy induced by *PANK2* OE in both WT and *PINK1* knockdown cells (Supplementary Fig. 6e, f). Similar to the genetic regulations, D-pan administration exhibited similar rescues of mitophagy in *PINK1* knockdown cells by promoting co-localization between autophagy markers (such as LC3, p62 and Lamp2) with mitochondria (Fig. 5i, j and Supplementary Fig. 6g–j). It also increased the degradation of autophagy receptors (p62, OPTN) and mitochondrial markers (Tom20 and Tim23), as well as LC3 lipidation, which meant that the mitophagy impaired by PINK1 deficiency was restored (Supplementary Fig. 6k, l). As controls, neither of *PANK2* OE and D-pan

administration would change the basal level of mitophagy in human cells without mitochondrial stress (e.g., CCCP treatment) in either immunoblotting (Supplementary Fig. 6m) or immuno-fluorescence staining experiments (Supplementary Fig. 6n–s). Moreover, a mitoKeima indicator was expressed in this human cell system, and we found that D-pan treatment significantly helped cells regain the mitophagy ability in *PINK1* knockdown cells (Supplementary Fig. 6t). Interestingly, RE-024, which is an alternative precursor bypassing PANK2 in CoA synthesis pathway, also significantly promoted mitophagy in *PINK1* knockdown human cells, as indicated by the colocalization between LC3 and mitochondrial marker (Supplementary Fig. 6u, v).

Conversely, *PANK2* knockdown in human cells after exposure to CCCP significantly inhibited mitophagy, manifesting as aloofness of autophagy markers (LC3, p62 and Lamp2) from mitochondria (Fig. 5k, l and Supplementary Fig. 7a–d) and the blockage of degradation of autophagy and mitochondrial markers (Supplementary Fig. 7e, f), while had no visible effect in non-treated cells (Supplementary Fig. 6m, 7g–i). In line with this, PKAN patient fibroblasts (loss of PANK2 activity) exhibited a paucity of mitophagy when stressed with CCCP. After 24 hours of CCCP treatment, PKAN fibroblasts could still maintain mitochondria compared to the control, and they also showed less LC3 lipidation at the basal level, indicating an impaired mitophagy ability (Supplementary Fig. 7j, k). In addition, the LC3 recruitment to damaged mitochondria was also largely diminished in PKAN fibroblasts after 4 hours treatment (Fig. 5m, n). These results suggest that the regulation of mitophagy by PANK2 and CoA metabolism is also conserved in human cells.

**Fbl promotes mitophagy *via* p62 acetylation**. To better understand the mitophagy process happened in these flies, we examined the sole autophagy receptor Ref(2)P and ubiquitin signals in fly muscle tissues. Prominent Ref(2)P puncta were present in the muscle cell cytoplasm of *PINK1* RNAi flies, while general diffuse ubiquitin signals were observed (Fig. 6a, b). Recruitment of ubiquitin to Ref(2)P was strongly reinforced by overexpression of Fbl, while reduced when Fbl was inhibited (Fig. 6a, b), suggesting a boost of Ref(2)P/p62-ubiquintin binding after enhancing Fbl activity. We also found an elevation of the Ref(2)P level in *PINK1* LOF flies (Supplementary Fig. 8a, b) in which it showed poly-ubiquitination. On this basis, *fbl* OE significantly strengthened it and vice versa (Fig. 6c). Intriguingly, the poly-ubiquitin signal could not be detected from the denaturing IP purified Ref(2)P protein from same samples, indicating that it is not from direct

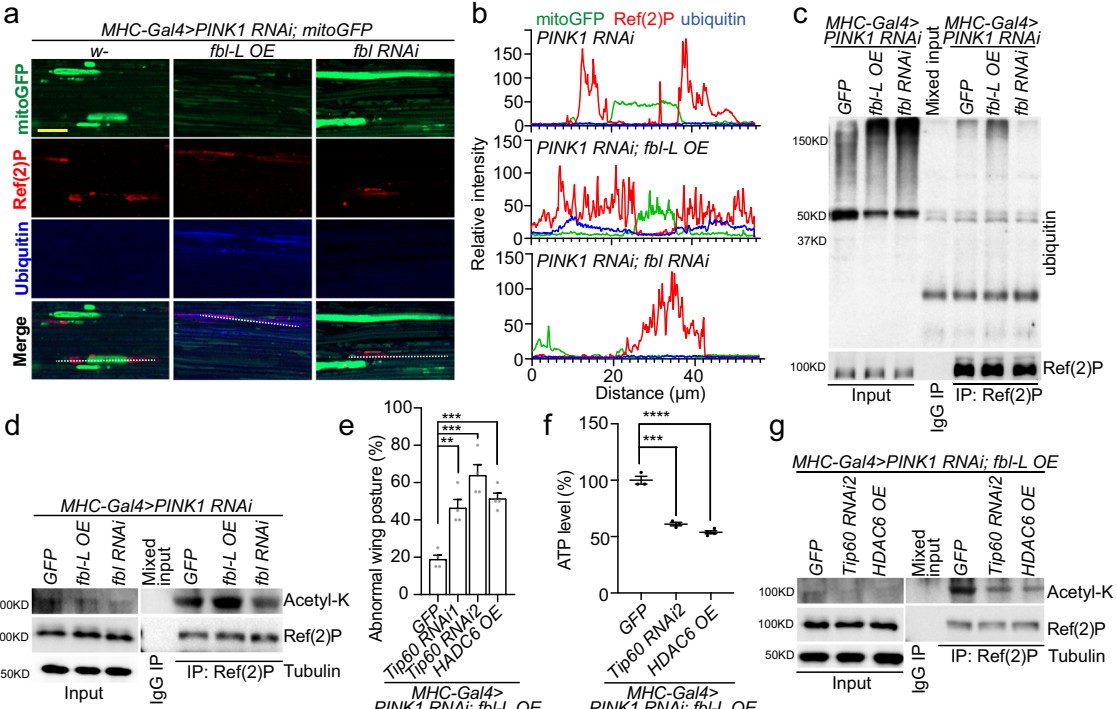

**Fig. 6 Fbl Augments Ref(2)P Acetylation to Promote Mitophagy. a** Immunostaining of ubiquitin and Ref(2)P puncta signals showing effect of Fbl in *PINK1* RNAi fly muscles. Scale bar, 25 μm. **b** Colocalization analysis of **a**. **c** Co-immunoprecipitation (co-IP) assay showing interaction of poly-ubiquitin signal and Ref(2)P regulated by *fbl* in *PINK1* RNAi flies. **d** Co-IP assays showing acetyl-lysine signal on Ref(2)P regulated by *fbl* in *PINK1* RNAi flies. For assays in (**c**, **d**, and **g**) *UAS-GFP* serves as control, and the rabbit IgG was used in control IP. **e** Aggravation of *Tip60* RNAi and *HDAC6* OE on wing posture defect showing their suppression of *fbl-L* OE in *PINK1* RNAi flies. 25 flies per genotype per group were tested; *n* = 4 biologically independent groups. **f** Effects of *Tip60* RNAi and *HDAC6* OE on ATP level showing their suppression of *fbl-L* OE in *PINK1* RNAi flies. *n* = 3 (*GFP*, *Tip60 RNAi2*), *n* = 4 (*HDAC6 OE*) biologically independent samples in assay (**f**) For all assays in (**e**, **f**) *UAS-GFP* serves as control. **g** Co-IP assays showing acetyl-lysine signal on Ref(2)P suppressed by *Tip60* RNAi and *HDAC6* OE in the background of *PINK1* RNAi *plus fbl-L* OE. For assays in (**e**, **f**) the significance was calculated by using two tailed unpaired *t*-test. Data are presented as mean values ± SEM; n.s. not statistically significant; *$p < 0.05$; **$p < 0.01$; ***$p < 0.001$; ****$p < 0.0001$. Source data are provided as a Source Data file.

ubiquitination of Ref(2)P itself, but from the other poly-ubiquitinated proteins bound to it (Supplementary Fig. 8c). Accumulating evidence suggested the importance of post-translational modifications of p62, such as acetylation, on its activation and binding ability with ubiquitylated proteins under nutrient stress[56]. While the general protein acetylation was not dramatically affected (Supplementary Fig. 8d), we still found increased acetylation on the immunoprecipitated Ref(2)P protein when *fbl-L* was overexpressed and declined signal when *fbl* expression was suppressed (Fig. 6d and Supplementary Fig. 8b).

Tip60 (Lysine acetyltransferase KAT 5) and HDAC6 (Histone deacetylase 6) have been respectively reported as the key acetyltransferase and deacetylase to balance the p62 acetylation[56]. To analyze their functions in Fbl-promoted mitophagy, we induced them in our *PINK1* LOF plus *fbl* OE combination. Interestingly, both *Tip60* RNAi and *HDAC6* OE remarkably suppressed the rescue of *fbl* OE in abnormal wing posture rate (Fig. 6e) and ATP production (Fig. 6f). Further, in more complete genetic analysis, we found that in *PINK1* RNAi flies, *Tip60* RNAi and *HDAC6* OE enhanced wing abnormalities, while *Tip60* OE and *HDAC6* RNAi significantly mitigated them (Supplementary Fig. 8e). Interestingly, *Tip60* RNAi and *HDAC6* OE acted more potently in *fbl* OE flies (*PINK1* RNAi background), whereas *Tip60* OE and *HDAC6* RNAi showed no further rescue, indicating that *Tip60* and *HDAC6* genes are downstream of *fbl* (Supplementary Fig. 8f). On the contrary, *fbl* RNAi nearly eliminated all actions of *Tip60* and *HDAC6* on *PINK1* RNAi flies, which can be explained by the consequence of severe reduction of CoA/acetyl-CoA, the

essential element for Tip60 and HDAC6 functions (Supplementary Fig. 8g).

To verify whether the regulation of Tip60 and HDAC6 on Fbl is through acetylation, we checked the acetylation of Ref(2)P in these flies. As expected, *Tip60* RNAi and *HDAC6* OE were found to obviously diminish the acetylation of Ref(2)P in flies (Fig. 6g), indicating certain similarity between human cell and fly models and in different subtypes of autophagy. In human cells, the K420 and K435 sites of p62 UBA domain are essential for mitophagy, as the p62 K2R (K420R, K435R, mimic of nonacetylated lysine residues) mutant loses the ability to colocalize with damaged mitochondria (Supplementary Fig. 8h, i). We subsequently performed in vitro acetylation assay on purified Ref(2)P by Tip60. Although they are not conserved in *Drosophila*, Ref(2)P still could be acetylated by Tip60 surprisingly, indicating the existence of other potential acetylation sites that can be recognized (Supplementary Fig. 8j). Lastly, we conducted an in vitro ubiquitin-binding assay to assess the contribution of acetylation on Ref(2)P to ubiquitin binding affinity. We found that acetylation by TIP60 on Ref(2)P facilitated its binding to ubiquitin (Supplementary Fig. 8k). In sum, we suggest that in *PINK1* LOF PD model, Fbl helps maintain the mitochondria integrity via acetylating/activating Ref(2)P, promoting ubiquitination and increasing mitophagy.

**fbl is locally translated on OMM and governed by PINK1/Parkin pathway.** A recent report alluded an increased level of pantothenate (i.e., Fbl/PANK2 substrate) in *PINK1* mutant; and

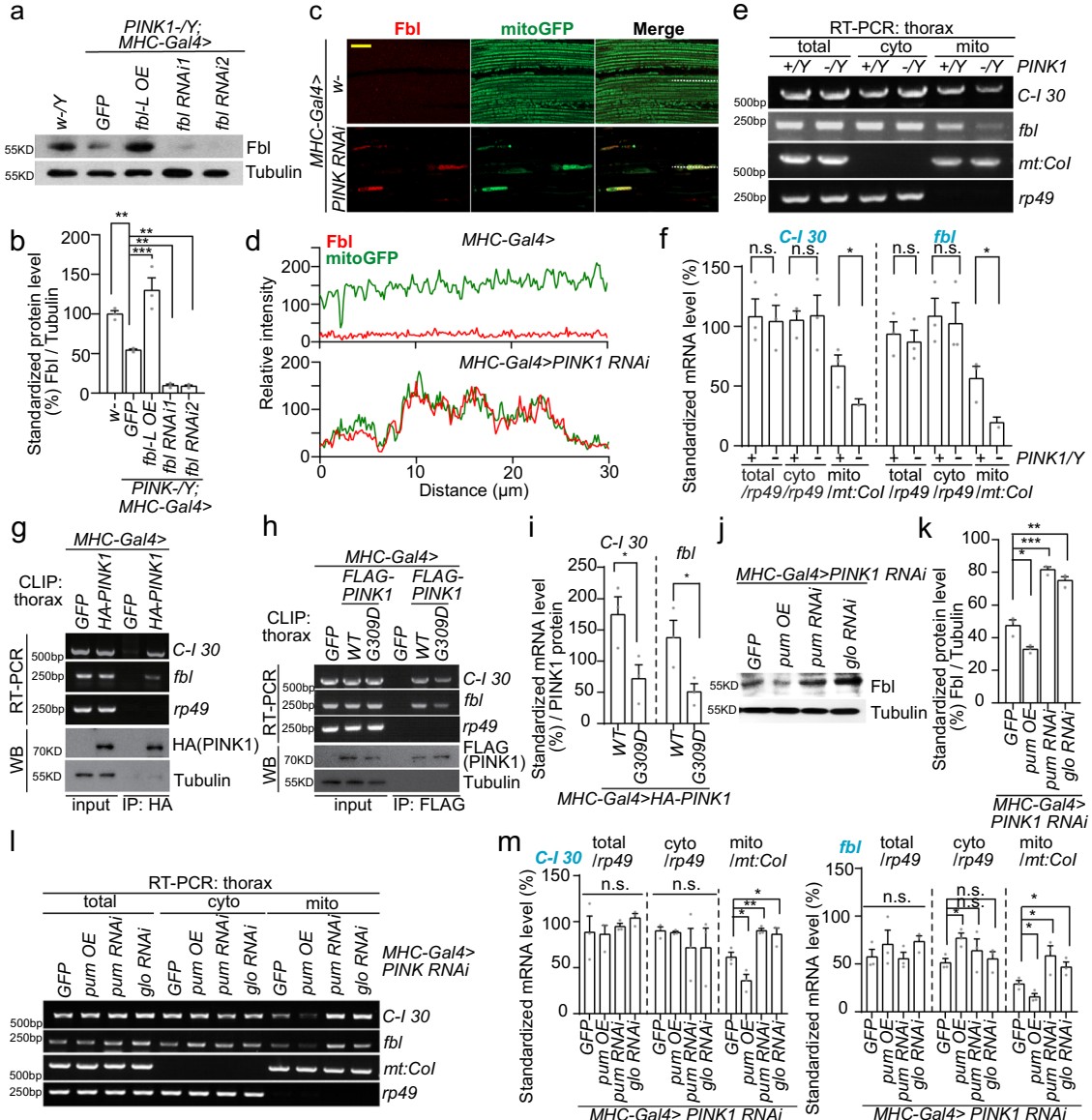

**Fig. 7 PINK1/Parkin Pathway Regulates *fbl* mRNA Translation on OMM. a** Immunoblot of Fbl protein levels in WT (*w-/Y*) and *PINK1* mutant showing the regulation by PINK1. Tubulin serves as loading control. **b** Quantification of **a** n = 3 biologically independent samples. In this and all subsequent immunoblot and RT-PCR quantifications, 3 independent biological repeats were measured per genotype. **c** Immunostaining of Fbl and mitoGFP signals showing the mitochondrial localization of Fbl in *PINK1* RNAi fly muscles. Scale bar, 25 μm. **d** Colocalization analysis of **c**. **e** Semiquantitative RT-PCR of *fbl* mRNA in *PINK1* mutant muscle. mRNAs prepared from total, cytosol (cyto), or mitochondrial (mito) fractions were used. Mitochondrial genome encoded gene *mt:CoI* serves as mitochondrial loading control; *rp49* serves as total and cytosol loading controls. **f** Quantification of **e**. For RT-PCRs in (**e** and **l**), n = 3 biologically independent samples. **g** CLIP assays showing PINK1 binding to *fbl* mRNA in fly. *C-I30* and *rp49* mRNAs served as positive and negative control, respectively. **h** CLIP assays showing reduced binding to *fbl* mRNA of PINK1[G309D] compared to PINK1 WT. **i**, Quantification of **h**, n = 3 biologically independent samples. **j** Immunoblot of Fbl protein levels showing the regulation of Pum and Glo. Tubulin serves as loading control. **k** Quantification of **j** n = 3 biologically independent samples. **l** RT-PCR showing effects of Pum and Glo on the defective mitochondrial localization of *fbl* mRNA in *PINK1* RNAi flies. *mt:CoI* serves as mitochondrial loading control, while *rp49* serves as total and cytosol loading controls. **m** Quantification of **l** n = 3 biologically independent samples. For assays in (**b**) the significance was calculated by using one way ANOVA followed by post hoc Dunnett's multiple comparisons test. For assays in (**f**, **i**, **k**, **m**) the significance was calculated by using two tailed unpaired *t*-test. Data are presented as mean values ± SEM; n.s. not statistically significant; *p < 0.05; **p < 0.01; ***p < 0.001. Source data are provided as a Source Data file.

in our preliminary assays, pantothenate also demonstrated no rescue to *PINK1* mutant (data not shown) as well as *fbl* mutant[57]. This observation implied a latent PANK dysfunction in *PINK1* LOF animals. When studying Fbl functions in these animals, we found Fbl level was suppressed in *PINK1* mutant and *PINK1* RNAi flies (Fig. 7a, b and Supplementary Fig 9a, b), and so was human PANK2 when *PINK1* was knocked down in human cells

(Supplementary Fig. 9c, d). Both *Drosophila* Fbl and human PANK2 are considered as the nuclear-encoded mitochondria targeted proteins[37,40], and recent studies have revealed a novel localized, mitochondrial membrane dependent, co-translational import mechanism of certain nuclear-encoded mitochondrial proteins on OMM facilitated by PINK1/Parkin pathway[35]. To study the mechanism of PINK1 controlling *fbl/PANK2*

translation, we first analyzed the Fbl localization in the fly muscle tissue when PINK1 was repressed. Unexpectedly, we found that although the total amount of Fbl protein was reduced, Fbl still colocalized with mitochondria, especially aggregated mitochondria in *PINK1* RNAi flies, suggesting that the aggregated mitochondria might be functionally more active (Fig. 7c, d). Notably, aggregated mitochondria showed a high membrane potential in TMRM staining indicating that aggregated mitochondria with high mitoGFP signal were formed in a compensatory protective way (Supplementary Fig. 9e). The data above suggest that Fbl could be the potential target of PINK1. We next analyzed *fbl* or human *PANK2* mRNA in the percoll-gradient purified mitochondrial fraction upon *PINK1* expression modulations. The amount of mRNAs localized to mitochondria was significantly lessened, with a corresponding increase in the cytosolic fraction, in *Drosophila PINK1* mutant or human *PINK1* knockdown cells (Fig. 7e, f and Supplementary Fig. 9f, g). We then conducted the crosslinking and immunoprecipitation (CLIP) assay to detect a possible direct interaction between *fbl* mRNA and PINK1[35]. Similar to the positive control fly *C-I30* (NADH dehydrogenase 30 kDa subunit) mRNA, *fbl* and human *PANK2* mRNAs were also found to robustly bind to PINK1 (Fig. 7g–i and Supplementary Fig. 9h–j). Intriguingly, mild mitochondrial stress (such as a short CCCP treatment) enhanced this interaction (Supplementary Fig. 9i), while a pathogenic form of PINK1 (PINK1$^{G309D}$) displayed a much lower binding affinity compared to WT PINK1 (Fig. 7h, i). In addition, we purified and analyzed *fbl* mRNA ribonucleoprotein (mRNP) complex by generating a construct expressing *fbl* mRNA tagged with MS2 binding sites and using the MS2-Tagged RNA Affinity Purification (MS2-TRAP) system[58,59]. Consistently, both endogenous, exogenous PINK1 and translational repressor Pumilio homolog 2 (Pum 2) were co-purified with the *fbl* and *C-I30* mRNPs in human cells (Supplementary Fig. 9k).

Translation repressors such as Pumilio (Pum; fly ortholog of human proteins Pumilio homolog 1, Pum 1 and Pumilio homolog 2, Pum 2), Glorund (Glo; fly homolog of heterogeneous nuclear ribonucleoprotein L, hRNP L) and heterogeneous nuclear ribonucleoprotein H (hRNP H) have been demonstrated having the ability to suppress the *C-I30* local translation process in multiple organisms[35]. For *fbl*, when knocking down *pum* and *glo* in *PINK1* RNAi flies, Fbl protein level could be restored significantly, while *pum* OE aggravated the decline (Fig. 7j, k). Consistent with the protein level changes, *pum* and *glo* knockdown effectively elevated *fbl* mRNA levels on mitochondria, whereas *pum* OE decreased it (Fig. 7l, m). We next tested the in vivo relevance of the above findings. Knocking-down translation repressors dramatically rescued *PINK1* RNAi defects as we showed before[35], however, *fbl* RNAi significantly suppressed their rescues (Supplementary Fig. 9l).

Next, we wanted to examine the regulation of *fbl* translation by Parkin since it works with PINK1 to regulate *C-I30* mRNA translation on OMM. Unsurprisingly, we found reduced Fbl levels and compromised *fbl* mRNA recruitments on OMM in *parkin* mutant flies (Supplementary Fig. 9m–p). More importantly, the regulation of *fbl* mRNA recruitment and translation was reflected in the synergistic effect of survival rate of *parkin* mutant and *fbl* RNAi flies, which led to lethality in the double mutant, as compared to only partial survival rate reduction in any single one (Supplementary Fig. 9q).

In summary, same as *C-I30*, *fbl/PANK2* mRNA also follows a localized translation mechanism. And as an important part of PINK1/Parkin pathway-related MQC when facing mild mitochondrial challenge, its translation is regulated by PINK1 and Parkin while largely repressed by Pum and Glo.

## Discussion

In this study, we established a sophisticated link between Fbl/PANK2, the first and speed-limiting regulatory enzyme in the CoA synthesis pathway and PINK1/Parkin, a central pathway in the MQC (Supplementary Fig. 10). *fbl* mutant displayed obvious mitochondrial dysfunctions in the central nervous system analogous to *PINK1* mutant, which was a recurrence of the commonalities of PKAN and PD. *fbl* OE rescued, while *fbl* RNAi aggravated, multiple defects associated with *Drosophila PINK1* mutant. Intracellular CoA and acetyl-CoA levels are largely influenced by Fbl/PANK2 bioavailability and they execute an important role in regulating mitophagy activity: Ref2p/p62 requires specific acetylation to facilitate its interaction with ubiquitinated proteins and initiate the downstream events in mitophagy. In the other half of the regulatory loop, reminiscent of certain nuclear-encoded mitochondrial RCC subunits[35], *fbl* translation follows a co-translational import mechanism on OMM and is tuned by PINK1, Parkin, Pum and Glo. Accordingly, PINK1-related PD should also suffer from Fbl/PANK2 deficiency. The stimulation of mitochondrial protein translation and import is recognized as an indispensable part of MQC controlled by PINK1 pathway to fight against mild mitochondrial damages. The MQC failure would cause more severe and broader mitochondrial dysfunctions in a positive-feedback manner due to the CoA shortage. These observations and underlying mechanism provide a rational explanation for overlapping symptoms of PD and PKAN as we discussed earlier.

We conclude that *fbl* is downstream of *PINK1* in this regulation. How could then the double knock-down of *PINK1* RNAi and *fbl* RNAi exhibit a worsening phenotype? From genetic points of view, clear genetic epistasis relationship can only be established with null mutations for loss of function of genes. Unfortunately, it is not possible to analyze this with null mutants; complete absence of Fbl would not be possible because CoA is such a vital metabolic intermediate molecule without which life cannot be sustained. The observation that *PINK1* RNAi was aggravated by *fbl* RNAi could be explained in two ways: (1) they may be in parallel pathways, working synergistically or together towards a specific phenotype; (2) they may be in a linear pathway, both of which, when knocked down, become limiting. In other words, aggravation of *PINK1* RNAi by *fbl* RNAi, this fact by itself, can only indicate they are interacting, but not enough to suggest they are in a linear pathway. To establish they are in a linear pathway needs more concrete evidence, such as *fbl* OE rescued *PINK1* mutant or RNAi (Fig. 2 and Supplementary Fig. 2), and PINK1 loss directly impacted *fbl* mRNA location and protein level (Fig. 7 and Supplementary Fig. 9). Considering that PINK1 loss affects *fbl* mRNA localization and expression, they would not be in parallel pathways. At molecular level, the explanation for *fbl* RNAi aggravating *PINK1* RNAi is as follows: *PINK1* RNAi reduced *fbl* mRNA positioning on the mitochondria, affecting its translation, and on top of this, *fbl* RNAi would make this worse. This explanation fits well with the above-mentioned scenario 2, where both are in a linear pathway and when knocked down, both are limited and in combination would worsen the phenotype. Consistent with this scenario, the Fbl protein level in *PINK1* and *fbl* duo RNAi (or mutant) fly was dramatically lower than that of *PINK1* RNAi alone (Fig. 7a and Supplementary Fig. 9a). Therefore, our data strongly supports that *fbl* is downstream of *PINK1*. Of course, Fbl, the enzyme catalyzing the key metabolic component CoA, does things more than its role in the PINK1-Fbl pathway.

Our genetic analyses indicate that Fbl/PANK2 works in parallel with Parkin, both downstream of PINK1. *fbl* OE could effectively rescue *PINK1* LOF but not *parkin* LOF in our tests. But on the other hand, biochemical analyses suggest that Parkin, just like

Pink1, is involved in Fbl translation. To better compare them, we summarized these congruent and disparate findings in a table (Supplementary Table 1). How could we understand the drastic differences in *fbl* rescue of the mutants of the two genes? We consider that differential roles of PINK1 and Parkin in regulating the level of cellular CoA and its derivant acetyl-CoA might be the key. Previous global analysis of metabolic changes in *Drosophila PINK1* null mutant revealed a downregulation of metabolites in CoA metabolism including CoA and acetyl-CoA, while pantothenate level is increased[36]. Consistently, our data indicate that PINK1 regulates the mRNA localization and protein level of *fbl*, leading to the defect of CoA and acetyl-CoA synthesis in *PINK1* mutant, which can be rescued by *fbl* OE. Unlike that in *PINK1* mutant (Fig. 2h, i), on the other hand, *fbl* OE could not restore the acetyl-CoA level in *parkin* mutant (Fig. 4i, j), which implies that a possible blockage exists in the conversion of acetyl-CoA from CoA when Parkin is affected. Intriguingly, both *parkin* OE and knockdown increased the CoA level in *PINK1* LOF, although further reduced acetyl-CoA level was found in the double knockdown flies (Supplementary Fig. 4d, e). The inability of *fbl* OE or D-Pan supplement to rescue Parkin LOF indicates genetically that *parkin* LOF is associated with a defect downstream of *fbl*, likely PDH, the critical enzymic protein complex catalyzing the conversion of pyruvate and CoA to acetyl-CoA. Indeed, there are reports showing that *PRKN* knockdown in H460 cells causes reduction of PDH complex activity and acetyl-CoA level drop[48]. We thus suspect that PDH dysfunction is also responsible for the CoA accumulation and acetyl-CoA decline in *parkin* mutant observed in our experiments. Encouragingly, regulation of *Pdha* effectively changed CoA and acetyl-CoA levels in *parkin* mutant (Supplementary Fig. 4a, b). Since the protein level of Pdha did not obviously change in *parkin* mutant (Supplementary Fig. 4c), our data suggest that the effect of Parkin on Pdha appears to be mainly through regulation of the enzymatic activity. Thus, back to the *PINK1, parkin* double LOF fly, the pathological accumulation of CoA and further depletion of acetyl-CoA may be related to the strong loss of PDH activity. This can be the main reason underlying the observation that *fbl* OE and D-pan rescue *PINK1* LOF flies, but not *parkin* LOF flies, since both interventions can only increase in vivo CoA levels but not mitigate the PDH complex defect resulted from *parkin* deficiency. In other words, *fbl* OE can enhance both CoA and acetyl-CoA levels in *PINK1* LOF, while only CoA but not acetyl-CoA in *parkin* LOF, explaining the differential rescuing effects of Fbl on *PINK1* and *parkin* LOF.

One important implication of this study is the likely accomplishable approaches for relevant disease therapies. In our study, D-pantethine (D-pan), fosmetpantotenate (RE-024) and 4'-phosphopantetheine (4'-PPT) were found improving multiple physiological indexes of *PINK1* mutant. Similar rescues of D-pan, RE-024 and 4'-PPT strongly indicate the rescue is via the CoA synthesis pathway. D-pan has been shown to have a positive effect in MPTP-induced PD rodent models[15]. Moreover, 4'-PPT and RE-024 were developed to treat PKAN[60,61]. Although in a recent clinical trial, FOsmetpantotenate Replacement Therapy (FORT) was shown not to be very effective[62]. In another single-arm, open-label study, pantethine was shown to delay the progression of motor dysfunction in children with PKAN, but not rescue[63]. Certain beneficial effects to PD are observed with therapies in our current studies. Since pathways of de novo synthesis, transport and metabolism of CoA, as well as relevant precursors are still not fully understood, differences between animal models (*Drosophila* and mouse) and humans may cause divergence in efficacy of treatment. It will be interesting to further confirm these effects in other systems such as PD patients' fibroblast-derived neurons. Another promising therapeutic

compound that was not included in this study, is kinetin, which has been proved to be a PINK1 activator[64,65]. In our study, Fbl/PANK2 expression is primarily determined by PINK1 activity. Kinetin treatment for mild mitochondrial damages might elevate the PINK1 activity, promote Fbl translation, augment CoA and acetyl-CoA production, boost mitophagy and help maintaining the mitochondrial health. Further evaluations can be performed in both PD and PKAN mouse models, while positive outcomes may lead to the development of improved disease therapies.

On the whole, mutations in the PKAN gene *PANK2* were not found in 67 familiar or early onset PD patients, nor in 339 idiopathic late onset PD patients[66], suggesting *PANK2* mutation is not a significant, if any, factor in PD. We suspect that *PINK1* mutations in PD result less severe decrement of PANK2 activity compared to *PANK2* mutations found in PKAN, since *PINK1* is upstream of *PANK2* and not the sole controller of its translational regulation. *PANK2* mutations can cause more direct and debilitating defects; PKAN patients often show symptoms at 3~4 years old and die in their early teens. It is also worth noting that PKAN belongs to a broader category of neurodegenerative diseases NBIA (Neurodegeneration with Brain Iron Accumulation). PKAN is known as NBIA1, and another subset include PLAN or NBIA2 (PARK14), mutated for *PLA2G6* (encoding PLA2G6, 85/88 kDa calcium-independent phospholipase A2), a gene involved in lipid metabolism[67,68]. NBIA2 exhibits widespread cortical alpha-synuclein-positive Lewy body pathology[69,70]. A far subset of NBIA is MPAN, mutated in a poorly characterized mitochondrial protein called C19orf12[71]. MPAN also has Lewy body-like, as well as tau-positive inclusions. These observations imply that NBIA as a whole also overlaps with PD, or they are more closely related than they appear[72].

Lastly, in this research, we conducted new customized statistics of mitochondrial aggregation/morphology in *Drosophila* muscle and DA neurons. We found the regular ONE-WAY ANOVA test or Student's t-test was not suitable in these scenarios, since their assumption are to test if the mean values of different groups have significant difference. The underlying biological hypothesis of assumption is that the genetic regulation will equally affect each mitochondrion. However, this is not what we observed in our samples; the aggregation randomly happens in a portion of the mitochondria population, leaving a large number of smaller individuals. This observation indicates that it may be more relevant to compare the behavior of the extreme values under different experiment conditions. The two-proportion Z test is used to determine whether percentages of values above a certain threshold are significantly different between the experiment group and the control. Thus, stepwise threshold screening can be utilized to determine specific thresholds that provide significance in these tests, and in addition, help us determine whether the modifications (genetic or pharmacological) will preferentially remove smaller or larger mitochondrial aggregates. The new tests are more in line with our observation and better reflect the changes in mitochondrial aggregates in our experiments.

## Methods

**Drosophila stocks**. Flies were normally raised at 25 °C, ~65% humidity on standard cornmeal media, unless otherwise noted. D-pantethine (Sigma) was added at a concentration of 0.8 mg/mL in the standard fly food.

Transgenic lines *UAS-Fbl-L* (Fbl-L OE) and *UAS-Fbl-S* (Fbl-S1 OE and Fbl-S2 OE) were generated in *w1118* background by P-element mediated transformation[38]. Fbl-L M2, M7 and yeast NDI1 lines were generated by injection of pUAST-Fbl-L (K221A) and pUAST-yNDI1 plasmids into *w1118* embryos. The expression of Fbl-L mutants (lines 2# and 7#) and NDI1 were examined by RT-PCRs. Some *fbl* RNAi lines were purchased from the Vienna *Drosophila* RNAi Center (VDRC, Vienna, Austria) (v101437 for *fbl* RNAi #1) and the Tsinghua Fly Center[73] (THU0131.N for *fbl* RNAi #2). UAS-ATG5 RNAi (THU2714), UAS-ATG8a RNAi (THU1555), UAS-ATG12 RNAi (THU2715) and UAS-Ppcdc RNAi (THU04283.N), were from Tsinghua Fly Center[73]. The *PINK1B9* mutant was a gift from Dr. Jongkeong

Chung[32,74]. The *TH-Gal4* line was a gift from Dr. Serge Birman. The *UAS-mitoGFP* fly line was a gift from Dr. William Saxton. *UAS-ref(2)P* lines were gifts from Dr. L M Martins[75]. The *UAS-PINK1 RNAi* line, *UAS-parkin RNAi* line, *UAS-parkin OE* line, *UAS-pum OE* line, *UAS-pum RNAi* line, *UAS-glo RNAi* line were gifts from Dr. Bingwei Lu[29,35]. The *UAS-Atg1 OE* line was a gift from Dr. Tao Wang at National Institute of Biological Sciences, Beijing. The *Parkin1* and *ParkinΔ21* mutants were provided by Dr. Patrik Verstreken[76]. All the other stocks were purchased from the Stock Centers (see Supplementary Table 2).

**Cell lines**. Mammalian cells were cultured under normal conditions (1x DMEM medium (Hyclone), 10% fetal bovine serum (FBS, Gibco), 5% $CO_2$, 37ºC, with supplementation of penicillin/streptomycin. Human fibroblast cells from PKAN patients (genotypes: c.36 G > T/c.979 A > G; c.1561 G > A/c.1502 T > C; c.1561 G > A/c.1561 G > A on *PANK2* genes) were from Prof. Susan J. Hayflick (Oregon Health and Science University, USA). The HEK293T cell was a gift from Prof. Yu Li Lab (Tsinghua University, China).

**Plasmids and molecular cloning**. The site-directed mutagenesis was performed to generate the kinase-dead form of Fbl-L (K221A: Lys 221→Ala)[40]. The DNA was cloned into pUAST vector via Xho-I/EcoR-I sites to generate the pUAST-Fbl-L (K221A) construct. The CDS sequence of *Drosophila Pdha* (*dPdha*) was cloned into pUOAST vector via EcoR1-XhoI sites to generate the pUAST-dPdha construct. The CDS sequence of human *PANK2* was cloned into pcDNA3.1 vector via Nhe-I/EcoR-I sites to generate pcDNA3.1-hPANK2 construct. The CDS sequences of *Drosophila ref(2)P* and *Tip60* was cloned into pMXB10 vector to generate pMXB10-Ref(2)P and pMXB10-Tip60. To generate the MS2-*fbl* plasmid, *fbl* 5'-UTR and 3'-UTR primers were used to amplify *fbl* 5'-UTR-CDS-3'-UTR cDNA and PCR products were cloned into pMS2 vector via Pme-I/Xho-I. MitoDsRed and mitoGreen plasmids were from Prof. Yangyan Wu at Hunan Agriculture University. The MitoKeima plasmid was from Dr. Hongguang Xia of Zhejiang University. The GFP-LC3 plasmid was a gift from Prof. Yixian Cui of Wuhan University. Flag-human p62, Flag-human p62(K2Q) and Flag-human p62(K2R) plasmids were obtained from Prof. Wei Liu of Zhejiang University. The HA-hPINK1 plasmid was from Guanghui Wang of Soochow University. The Lamp1-RFP plasmid was from Li Yu of Tsinghua University.

**Compounds**. The fosmetpantotenate (RE-024) was prepared according to the protocol as reported[77]. 4'-phosphopantetheine (4'-PPT) was provided by Prof. Susan J. Hayflick (Oregon Health and Science University, USA).

**RNA purification, RT-PCR, and CLIP assay**. RNAs were extracted from different subcellular fractions by following the manufacturer's instructions from TRIzol Reagent (Invitrogen) or RNeasy Kit (Qiagen). At least 10–15 flies or fly thoraces were used for RNA extraction. cDNAs were synthesized using the TransScript one-step gDNA Removal and cDNA Synthesis Super Mix kit (TransGen Biotech), and RT-PCRs were performed using RT-PCR kit (Qiagen) with the corresponded primers. PCR products were mixed with 6x DNA loading buffer and run on 1.2–1.5% Agrose (Invitrogen) gels; signals were analyzed by Tanon 1600 UV Gel Imaging System.

The Cross-linking and immunoprecipitation (CLIP) assay was performed using a modified method. Briefly, cells or fly lysates were in the ice-cold 1x PBS while being irradiated once at 400 mJ/cm[2] using a Stratalinker (12 cm distance from UV source, Stratagen model 2400). Subsequently, cells were mixed with lysis buffer containing 1% Triton X-100. Protein of interest were immunoprecipitated for 3 h at 4 °C and followed by three washing steps, 5 minutes each, with 1x PBS/1% Triton X-100/0.05% SDS at 4 °C. Later, protein/RNA complexes were mixed with SDS sample buffer, run on 4–12% SDS-PAGE gels, and transferred onto PVDF membranes. A small membrane strip was cut at the expected sizes of the immunoprecipitated protein. A 4 mg/ml proteinase K solution in 1x PBS was pre-incubated at 37 °C for 20 min to digest away any contaminated RNases. Subsequently, 30 ul of the preincubated proteinase K solution was incubated with the membrane at 37 °C for 20 min, followed by RNA extraction using RNeasy® Mini kit (Qiagen) and RT-PCR analyses as described above.

**Immunoblotting**. For immunoblotting analysis of fly samples, male flies at 5~7 days age at 29 °C were used. 15 flies or 15 fly thoraxes were directly homogenized in lysis buffer [50 mM Tris-HCl pH7.4, 150 mM NaCl, 5 mM EDTA, 10% glycerol, 1% Triton X-100, 0.5 mM DTT, 60 mM β-glycerolphosphate, 1 mM sodium vanadate, 20 mM NaF, Complete protease inhibitor cocktail (Roche)], samples were centrifuged at 12,000 g and mixed with 5XSDS-loading buffer, separated on 12% SDS-PAGE, and then transferred to Immobilon P^SQ PVDF Membranes. Membranes were hybridized with corresponding primary antibodies at 1:1000 dilution (see Supplementary Table 2) for overnight at 4 °C, secondary antibodies at 1:10,000 dilution and then analyzed with SuperSignal™ West Dura Extended Duration Substrate kit (Thermo Scientific).

**Blue native gel analysis**. Blue native gel electrophoresis reagents were purchased from Invitrogen and electrophoresis analysis was performed based on the

manufacture's recommended protocol as described[78]. The purified mitochondrial samples were solubilized by 5% Digitonin (Invitrogen) on ice for 30 min and prepared as samples for Blue Native PAGE using the NativePAGE™ Sample PreP Kit (Invitrogen).

**Longevity assay**. The longevity assay was performed as described previously, with slight modifications[79]. Newly hatched flies were collected and separated randomly into different groups and transferred to fresh food every 2 days after being counted and recorded. 3–4 parallel groups with a total number over 100 individuals were used for each genotype.

**Abnormal wing posture and jumping/flight activity analysis**. To analyze abnormal wing posture and jumping/flight activity, male flies were aged at 29 °C with 20~25 flies (wing posture) and 10 flies (jumping and flight) per vial. The penetrance of abnormal wing posture was calculated as the percentage of flies with either drooped or held-up wing posture[74]. For each experiment, at least 60 flies were scored for each genotype. The jump/flight activity analysis was performed as previously described[29,74] with 10 flies per vial. Each of these experiments had been repeated at least three times.

**Mitochondrial morphology and immunohistochemical analysis**. The mitoGFP protein was expressed in *Drosophila* muscle (by *MHC-Gal4*) and dopaminergic neurons (by *TH-Gal4*) to indicate mitochondria morphology. To analyze mitochondrial morphology in muscle, fly thoraxes were fixed in PBSTx (1x PBS, 0.25% Triton X-100) with 4% paraformaldehyde, dissected and examined under confocal fluorescence microscope (Zeiss). To analyze mitochondrial morphology in dopaminergic neurons, *Drosophila* brains were fixed, dissected, stained, and mounted following standard procedures[76]. Primary antibodies used here were rabbit anti-TH (1:500/29), rabbit anti-ATG8 (1:1000), and mouse anti-ATP5a (1:1000). All confocal images were taken with a Zeiss LSM710 Meta confocal microscope and analyzed by the ZEN lite Digital Imaging Software.

**Mitochondria purification**. Crude mitochondria fractions were isolated from 3rd instar larvae as described[44]. Briefly, 3rd instar larvae were gently homogenized in cold mitochondrial isolation buffer (HBS buffer: 5 mM HEPES, 70 mM Sucrose, 210 mM mannitol, 1 mM EGTA, 1x protease inhibitor cocktail). The homogenate was then centrifuged at 1000 g for 10 min twice at 4 °C to remove the pellet. The supernatant was further centrifuged at 10,000 g for 15 min twice, and the pellet containing mitochondria was resuspended in the isolation buffer. The mitochondrial protein concentration was measured by the BCA kit (Thermo Scientific).

Intact mitochondria were isolated using the previously described methods[80]. Briefly, flies at appropriate ages were used for thoracic sample collection. To block the release of mRNAs associated with mitochondria, 0.1 mg/mL cycloheximide was applied to all buffer solutions. Samples were homogenized using a Dounce homogenizer. After two steps of centrifugation (1500 g and 13,000 g), the mitochondria pellet was washed twice with HBS buffer, then resuspended and loaded onto Percoll gradients (15%). After 16,700 g 15 min centrifugation, the fraction between the 22% and 50% Percoll gradients containing intact mitochondria was carefully transferred into a new reaction tube, mixed with HBS buffer, and centrifuged again at 22,000 g for 30 min at 4 °C to collect the pellets for further analyses[35].

**Immunoprecipitation and mRNP purification**. For immunoprecipitation, extracts of fly tissues were prepared by homogenization in lysis buffer. After centrifugation at 10,000 g for 5 min, the supernatant was subjected to immunoprecipitation using the indicated antibodies and protein A/G magnetic bead (Pierce) at 4 °C for overnight. Immunocomplexes were analyzed by SDS-PAGE and immunoblotting.

For denaturing immunoprecipitation, 30 fly thoraces of each genotype were collected and homogenized with 500 μl cell lysis buffer; samples were mixed with 500 μl 2X SDS Laemmli loading buffer supplemented with 2'-mercaptoethanol and denatured by boiling at 100 °C for 10 min. Then samples were diluted with pre-chilled cell lysis buffer at a 1:5 ratio and subjected to immunoprecipitation by using rabbit anti-Ref(2)P (Abcam).

For Fbl and C-I 30 mRNP purification, pcDNA3.1-PINK1-FLAG, MS2-BP-GST, and MS2-bs-C-I 30 or MS2-bs-Fbl plasmids were transiently expressed together. Sixty hours post-transfection, we UV crosslinked attached cells in the petri dish and purified mitochondria as described[36]. We homogenized the mitochondria fraction in the lysis buffer and performed GST pull-down assays using glutathione-sepharose beads (BeaverBio) at 4 °C for 6 h with gentle shaking. Subsequently, beads were washed three times (10 min each) at 4 °C in lysis buffer, mixed with 2x SDS Sample buffer, and loaded onto SDS-PAGE gels.

**shRNA and siRNA transfection**. human *PINK1* shRNAs were purchased from human shRNA library in Tsinghua University (#TRCN0000199193, #TRCN0000199446), and packaged by the lentiviral system (vector, packaging, and envelope plasmids with the plasmid proportion 3:2:2) by using P2PAX2, PMD2G plasmids and the HEK293T cell[81]. Lentiviruses were harvested in 36 h and transfected into HEK293T cells with 8 μg/ml Polybrene, after 24 h, lentivirus contained

medium was exchanged to new DMEM medium with 10%FBS, 1 µg/ml puromycin was supplemented after 48 h, protein extract from the transfected cells was used to analyze human PINK1 protein level. Human PANK2 was knocked down by siR-NAs (Sangon Biotech), which was transfected by Attractene Transfection Reagent (Qiagen).

**Analyzing mitophagy activity in mammalian cells.** To analyze the mitophagy activity in WT and *PINK1* RNAi cells, mitochondrial and autophagy markers were expressed in HEK cells as follows: mitoDsRed/GFP-LC3, mitoDsRed/FLAG-p62 and mitoGreen/Lamp1-RFP. Transfected cell lines were selected by treating with 200 µg/ml puromycin. Cells were subsequently treated with CCCP for 12 h, and the colocalizations of mitochondrial and autophagy markers were analyzed by confocal microscopy after immunofluorescence staining. To study the effect of CoA precursors on mitophagy, 100 ng/ml D-Pan and 300 ng/ml RE-024 were administrated 12 h before CCCP treatment. Cells with the same treatments were also harvested and subjected to subsequent SDS-PAGE and immunoblotting analysis.

To analyze the mitoKeima signal in living cells, HEK293 cells containing mitoKeima marker were treated with CCCP for 12 h. The live cell imaging was captured and analyzed by a Zeiss LSM880 confocal microscope via two sequential excitations as described[82].

**Mito-lysosome analysis by mitoQC.** Fly thoracic muscle tissues were isolated, fixed in PBSTx (1x PBS, 0.25% Triton X-100) with 4% paraformaldehyde, dissected and examined directly under confocal fluorescence microscope (Zeiss). A square area of 15.6 µm × 15.6 µm (200 pixels × 200 pixels) was selected for mito-lysosome analysis, and 15 samples from 3 independent biological replicates were counted.

**Measurement of mitochondrial oxygen consumption and complex activity.** The oxygen consumption of isolated mitochondria was measured at 25 °C as described in the respiratory buffer (120 mM KCl, 5 mM $K_2HPO_4$, 3 mM HEPES, 1 mM EGTA, 1 mM $MgCl_2$, and 0.2% fatty acid free BSA)[44,83]. 5 mM L-proline and 5 mM pyruvate or 7 mM succinate plus 5 µM rotenone were added to the respiration buffer as substrates to drive the Complex-I or Complex-II dependent respiration, respectively. The complex-III dependent respiration was measured with 20 mM sn-glycerol 3-phosphate as the substrate[84]. The activity of mitochondrial complex-I was measured by spectrophotometric assays (Thermo MLTISCAN G0) as described[85]: 1 mM NADH was used as complex-I substrate and spectrophotometric change at 340 nm was counted for 3 min.

**ATP measurement.** The thoracic ATP level was measured using a luciferase based bioluminescence assay (ATP Bioluminescence Assay Kit HS II, Roche) as described[74]. For each measurement, five muscle thoraces dissected from 7-day-old male flies were collected. The muscle thoraces were homogenized in 100 µl lysis buffer, boiled for 5 min, and centrifuged at 20,000 g for 3 min. 2.5 µl of the supernatant was mixed with 187.5 µl of dilution buffer and 10 µl luciferase/luciferin mixture provided with the kit. The luminescence was immediately measured using a luminometer (Promega, USA) or microplate reader (BioTek, USA). For each sample at least three independent measurements were performed.

**Coenzyme A and Acetyl-Coenzyme A measurements.** Levels of Coenzyme A and acetyl-Coenzyme A were assayed by kits (Sigma) according to the manufacturer's instructions. In HPLC analyses of CoA and Acetyl-CoA levels, fly larvae, adults or thoracic samples were collected and quenched by liquid nitrogen, and homogenized in methanol buffer (Methanol:dd$H_2O$ = 8:2) on dry ice. The homogenate was stored at −80 °C for 12 h, and then centrifuged at 12,000 rpm, 4 °C for 10 min. The supernatant then was collected and loaded for HPLC assay (Tsinghua Biomedical Centre). Contents of CoA and Acetyl-CoA were normalized by the sample weights.

**In vitro acetylation assay and in vitro ubiquitin binding.** *Drosophila* Ref(2)P and Tip60 proteins were purified from *E. coli* BL21 strain by using the chitin beads (NEB). The protein buffer was exchanged to (Histone Acetyltransferase Assays) HAT assay buffer (50 mM Tris-Hcl pH 8.0, 10% glycerol, 0.1 mM EDTA, 1 mM dithiothreitol)56, by using the Zeba™ Spin Desalting Columns (Thermo Scientific). In vitro acetylation assay was performed by incubated Ref(2)p (substrate) and Tip60 (enzyme) in HAT buffer with 1 mM Acetyl-CoA (Shanghai Yuanye Bio-Technology) at 30 °C for 4 h. The acetylation on Ref(2)P was then analyzed by immunoblotting with pan Acetyl-K antibody (Abcam).

For in vitro ubiquitin-binding assay, the Ref(2)P protein was expressed in *E.coli*, then purified and conjugated to chitin-beads. In vitro acetylation was performed as described above, and Ref(2)P-chitin beads were subsequently rinsed for 3 times with modified ubiquitin-binding buffer[86] (50 mM HEPES pH 7.5, 150 mM NaCl, 1.5 mM $MgCl_2$, 1 mM EGTA, 1% Triton X-100, 10% glycerol, 1 mM PMSF, 1 mM sodium vanadate with the protease inhibitor cocktail). Later, the ubiquitin-buffer with 2 µg ubiquitin (Gentihold) was provided and incubated for 2 h, and then rinsed for three times with new ubiquitin-binding buffer. After that, Ref(2)P

proteins on beads were collected and analyzed by immunoblotting using the anti-ubiquitin antibody described as above.

**TMRM staining in *Drosophila* muscle tissue.** Fly thoraces were collected by removing heads, wings, legs, and abdomen from whole flies and dissected in the Schneider's Medium (GIBCO™). Samples were washed once with fresh medium and stained in Schneider's medium (Dye concentration: TMRM 200 nM) at room temperature for 30 min in the dark room. After staining, samples were washed 3 times with the Schneider's Medium and directly observed by the Zeiss LSM710 Meta confocal microscope and analyzed by the ZEN lite Digital Imaging Software.

**Signal quantifications and colocalization analysis.** All immunoblot signals were quantified by using Image Lab (v6.0.1, Bio-Rad). ImageJ (v1.53k) was used for mitochondria quantification in *Drosophila* muscle and neurons[87]. The background was first removed, then sizes of every single mitochondrion over threshold (for muscle samples, >0.5 µm²; for neuron samples, >0.1 µm²) were recorded. Typically, three individual figures in each group were analyzed. For colocalization analysis, the Zeiss profile function (ZEN3.1, blue edition) was used.

**Reproducibility and statistics.** Representative images of CLIP assays, co-IPs, immunoblots, blue native gel analyses, mitochondria analyses in *Drosophila* tissues and immunofluorescence staining in both *Drosophila* tissues and human cells shown were replicated 3 times independently unless otherwise stated in figure legends.

All analyses were performed with Graphpad Prism 9.2.0 software, except for the two-proportion Z test, which was programmed and calculated by R (v4.1.1) language (Figs. 1b, 2d, f, 3f, 4d, f, Supplementary Fig. 2c, threshold was set to 3 µm²; Fig. 3d, threshold was set to 8 µm²; Supplementary Fig. 3o, threshold was set to 10 µm².). For pairwise comparisons (Figs. 2h–k, 3a, b, g–i, 4a, b, g–j, 5b, e, h, j, l, 6e, f, 7f, i, k, m, n, and Supplementary Fig. 1b, d, 2e–g, i–l, 3a–i, k, m, 4a, b, d–f, 5a, d, f, h, j, 6b, d, f, h, j, l, v, 7b, d, f, k, 8i, 9b, d, g, j, l, n, p, q) two-tailed Student's *t* test were used, confident interval is 95%. Data are presented as mean values ± SEM (the Standard Error of the Mean). For comparing multiple groups, we used one-way ANOVA test followed by Sidak's multiple comparisons test (Figs. 2g, 5a, and Supplementary Fig. 4e) or Dunnett's multiple comparisons test (Figs. 1c–h, 2a, b, g, 5f, 7b, and Supplementary Fig. 1f, g, 2a, d, 8e–g). For lifespan analyses, Log-rank (Mantel–Cox) test was used (Fig. 4l and Supplementary Fig. 2n, 3l, 4g). For the non-inflation wing ration, Fisher's exact test was used (Supplementary Fig. 1h). For the inflation wing ratio, Chi-square test was performed (Supplementary Fig. 5m, df = 13.74, 7). In our statistical comparisons, *$p < 0.05$, **$p < 0.01$, ***$p < 0.001$, and ****$p < 0.0001$. Sample size/statistical details were stated separately in the figure legends.

**Description of two-proportion Z test.** All the mitochondrial (mitoGFP) signals (area) were quantified by ImageJ automatically and subjected to following two-proportion Z test. The two-proportion Z test was used to compare two observed proportions. We used it to determine whether the percentages of values above a certain threshold are significantly different between the experiment group and the control. The null hypothesis and alternative hypothesis of the test are:

$$H_0 : p_e - p_c = 0; H_1 : p_e - p_c \neq 0 \qquad (1)$$

where $p_c$ *and* $p_e$ are defined as the proportion of values above the pre-specified threshold in the control group and the experiment group, respectively. The null hypothesis assumes that the proportions are the same in both groups, i.e., the treatment is not distinct from the control in terms of the behavior of the extreme values. The alternative hypothesis assumes these proportions differ, i.e., the behavior of the extreme values in the treatment differs from that in the control.

The test statistic (Z-score) is defined as:

$$Z = \frac{\hat{p}_e - \hat{p}_c}{\sqrt{\hat{p}_0 (1 - \hat{p}_0) \left( \frac{1}{n_e} + \frac{1}{n_c} \right)}} \qquad (2)$$

where $\hat{p}_0$ is the overall proportion which is calculated as portion of observed values above the threshold across the two groups, $n_c$ and $n_e$ are the sample size in the control group and the experiment group, respectively. The Z-score follows a standard normal distribution under the null hypothesis, the p-value is calculated accordingly.

**Reporting summary.** Further information on research design is available in the Nature Research Reporting Summary linked to this article.

## Data availability

All source data (raw images and statistics) used and reported in this study are provided in the Source Data file.

## Code availability

The code for Z-test used in this study are provided in the Supplementary Information.

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

## Acknowledgements

This research was supported by the National Basic Research Program of China (2018YFA0900100 to B. Zhou), the National Science Foundation of China (31971087, 91649118 to B. Zhou, 31700883 to Y. Huang, and 31700881 to Z. Wan), the National Institutes of Health (R15AG067470 to Z. Wu, and R01NS084412 to B. Lu), the Cancer Prevention and Research Institute of Texas (RP210068 to Z. Wu), the Children's Brain Diseases Foundation (411903 to Z. Wu) and Southern Methodist University (new faculty startup fund to Z. Wu). We are grateful to the Bloomington Stock Center, the Tsinghua Fly Center, and the Vienna *Drosophila* RNAi Center for fly stocks. We are indebted to Dr. O. Sibon for the anti-Fbl rabbit polyclonal antibody. We thank our friends and other research labs mentioned in the manuscript for their generosity in providing us relevant fly stocks and reagents. We also thank Dr. Johnson Taylor for editing and suggestions. The Biomedical Analysis Center of Tsinghua University provided technical help and services.

## Author contributions

B.Z., Z.Wu, Y.H., Z.Wan, and Y.T. designed the experiments. Y.H., Z.Wan, and Y.T. performed the experiments, analyzed data, wrote the manuscript, and contributed equally to this project. C.Y., B.Liu, and J.C. performed the Two-Proportion Z Test. J.X., B.Laboret, and S.N. helped with some of the fly experiments. B.Lu, Y.H., R.L., J.F., and S.H. provided key reagents and advice. B.Z. and Z.Wu conceived and supervised the study and wrote the manuscript. All authors revised and approved the article.

## Competing interests

The authors declare no competing interests.
