## [Peer Review File · Nature Communications]

Pantothenate Kinase 2 Interacts with PINK1 to Regulate Mitochondrial Quality Control via acetyl-CoA metabolismREVIEWER COMMENTS

Reviewer #1 (Remarks to the Author):

This is a very interesting paper showing a potential functional correlation between PINK1 and mitochondrial protein Pantothenate Kinase 2 in the regulation of mitochondrial quality control.

The topic is relevant and can potentially interest many scientists working in this field. By using fruit fly genetic as in vivo approach, the authors demonstrate a genetic interaction between mitochondrial quality control protein PINK1 and fumble (homologue of PANK2), a key enzyme in CoA metabolism. Starting from the observation that fumble mutants phenocopy PINK1, and that overexpressing fumble in PINK1 KO background rescues PINK1 KO phenotype (but not Parkin), authors conclude that the two proteins genetically interact, with PINK1 being upstream of fumble in a pathway that regulate MQC. In the proposed mechanism, Fbl regulates the acetylation of p62, thus directly promoting p62-dependent mitophagy. In addition, PINK1 controls mitochondrial anchoring and import of fbl mRNA, which is required for translation of this nuclear encoded enzyme so important for CoA synthesis.

In my opinion, the fact that PINK1 affects fbl mRNA anchoring and its mitochondrial import, is per se very interesting and deserves publication. The role of fbl in the regulation of MQC is a nice addition to a very intriguing story, but as it is, the study is too preliminary for publication. I am particularly concerned about the analysis of mitophagy, the biochemical nature of Fbl-p62 interaction, and the interpretation of the epistatic studies. Please see below for details:

(i) The epistatic analysis of PINK1-Fbl interaction is puzzling: in the double mutant (PINK1 RNAi and Fbl RNAi), authors show a significant aggravation of the phenotype which shouldn't be the case if, as proposed by the authors, Fbl and PINK1 operate in a linear pathway with PINK1 upstream of Fbl. This result actually indicates that more likely Fbl affects other pathways that are independent of PINK1 and MQC, and presumably impacts mitochondrial metabolism.

(ii) Mitophagy investigation is underdeveloped. Analysis of the involvement of autophagy genes (ATG1 and Ref2p/p62) is not a read out for mitophagy. The in vivo studies proposed in fig.4 are indeed interesting and indicative of an involvement of autophagy in the mechanism of function of Fbl. Mitophagy is however a selective type of autophagy that involves the degradation of mitochondria. There are no evidences in this paper that mitochondrial degradation by autophagy (i.e. mitophagy) is differentially affected by Fbl, and in combination with PINK1. Accumulation of p62 and/or LC3/ATG8 - decorated mitochondria cannot be interpreted as an enhancement of mitophagy, as this could be the result of impaired flux (i.e. impaired degradation). Authors should clearly show increased number of mitochondria and/or mitochondrial parts undergoing degradation in this condition, paralleled by decreased mitochondrial mass (in case mitochondrial biogenesis is not affected, this should be also tested). In flies mitochondrial degradation by the lysosomes can be specifically addressed in vivo by

using mt-QC or mt-Keima previously generated for the characterization of MQC in vivo in flies (please see for example Lee JJ et al., The Journal of cell biology 2018).

(iii) It is unclear how Fbl promotes p62 acetylation. Authors suggest a potential involvement for acetyltransferase and deacetylase Tip60 and HDAC6 but how the activity of these enzymes correlates to Fbl function is completely unexplored. The nature of this interaction could be direct or indirect and deserves in deep investigation.

Reviewer #2 (Remarks to the Author):

While rare, Pantothenate Kinase-Associated Neurodegeneration is the most common form of neurodegeneration with brain iron accumulation (NBIA) and is caused by loss of PANK2 function. PANK2 (or fumble/fbl in flies) encodes a mitochondrial form of pantothenate kinases, which catalyze the first/key step of CoA synthesis. Here the authors quite convincingly show a genetic interaction between PANK2 and PINK1 homologs in flies. PINK1 and Parkin together mediate a conserved neuroprotective mitochondrial autophagy pathway, but are both mutated and lost in early-onset Parkinson's disease. Fbl and dPINK1 mutants/RNAi display similar morphologic and functional mitochondrial abnormalities (at least in neurons), and overexpression of the mitochondrial, but not cytosolic (or kinase dead) Fbl isoform partially restored phenotypes of PINK1 loss of function, whereas Fbl RNAi exacerbates these. Likewise, d-pantethine supplementation restores CoA levels and is able to partially rescue PINK1 phenotypes as well. This is a quite solid part of the study, but the mechanistic aspects are very limited and less convincing and will need substantial work to be backed up. This includes both additional biochemical and immunofluorescence validation and adequate quantification of the respective effects. Moreover, it will be essential to clearly demonstrate conservation of the effects and mechanisms beyond drosophila, especially given that the authors try to link this to human disease. Further it is quite puzzling that fbl/PANK2 seem to operate downstream of PINK1, but in parallel to Parkin, although it is then implicated in pathways that are regulated by both, PINK1 and Parkin, such as mitophagy or the localized mitochondrial translation of select mRNAs. This needs to be substantiated too.

Overall this is an interesting manuscript with novel findings and potential disease relevance, but I have several major concerns that need to be addressed:

- 1) mitochondrial aggregation phenotype (labeled by mitoGFP)

This phenotype is actually never quantified, so here we rely on single immunofluorescence images for both TH neurons and muscles. A more objective measure such as of number/size of the aggregates or GFP intensity is needed to back up claims that these aggregates are “prominent” or “correlate with..”.

Also Fig 1a clearly shows similar aggregates for the Fbl-L OE strain although it is stated: ‘Prominent mitochondrial aggregations (labeled by mitoGFP) were observed in the DA neurons of Fbl RNAi flies, reminiscent of that in dPINK1 mutant (Fig. 1a), while Fbl-L OE did not generate any notable abnormality (Fig. 1a).’

2) Complex I

How do the authors explain a reduced complex I dependent respiration, but no reduction of complex I activity? Please explain including the differences between PINK1 and fbl phenotypes and rescue regarding complex I.

3) Parkin

Despite the clear genetic interaction between PANK2 and PINK1, lack of the same for Parkin is very puzzling and needs further experimental validation and additional discussion. Although it appears that neither Fbl overexpression, nor d-pantethine supplementation rescued parkin KO phenotypes, several questions remain some of which are:

Does fbl RNAi exacerbates Parkin KO phenotypes, similar to a PINK1 mutant background?

Does Parkin OE restore the acetyl/CoA levels in PINK1 KO animals?

What are the acetyl/CoA levels in PINK1/Parkin double mutants?

Also this investigation should be expanded to manipulation p62 and Atg genes in Parkin KOs

Are PANK2 levels affected in Parkin mutant flies/cells?

4) decline in mitophagy:

The authors actually never measure mitophagy, but rely on western blot readouts or single immunofluorescent images. Neither of these can inform on mitophagy which is a dynamic process and must be measured as such. Also none of that data shown for p62, ubiquitin or LC3 has been quantified, but all findings need to be corroborated both biochemically and by immunofluorescence to corroborate the findings in flies and human cells.

How do authors explain the lack of genetic interactions with parkin in the context of mitophagy?

5) human cells.

Fibroblasts are only N=1 for each mutant and controls. Data needs to be shown on the same western blot and needs quantification for each genotype and treatment. If there are no additional samples, this at least needs technical repeats and a more thorough investigation over time.

Anything with PINK1 or knockdown in human cells needs to be controlled by +/- CCCP treatment and blotting for endogenous PINK1, Parkin, and phosphorylated ubiquitin which are all determinant of mitophagy. Also PANK2 protein levels need to be shown.

Ext Fig 6C: There really shouldn't be any endogenous full-length PINK1 protein detectable in unstressed human cells (input).

6) P62 acetylation:

It is highly likely that there is an effect on overall protein acetylation and that this is not exclusively specific to p62. This line of research needs to be substantiated by other means and must be expanded to include changes in total cellular acetyl-lysine levels as well as other known controls such as tubulin and histones. In order to confirm a specific effect of acetyl-p62, for instance acetylation-mimic and -deficient version of p62 can be studied.

7) RNA:

The mRNA levels in different fractions needs to be shown on the same blot/gels alongside proper positive and negative controls and need quantification by qPCR. Likewise, CLIP needs to be quantified when comparing PINK1 WT and mutant, relative to the amount of IP'ed PINK1 protein.

Also in this context, how do authors explain their Parkin findings as both PINK1 and Parkin were suggested to coordinately derepress translation of OXPHOS-related mRNAs by targeting the Glo for ubiquitination.

Minor:

"Dramatic decline of complex I-, II- and III- dependent respiration was observed in Fbl LOF flies (Fig. 1d-1f), also similar to that in the dPINK1 LOF fly." This looks like an approximately 25-50% reduction, so mild to moderate decrease would perhaps be more appropriate than "dramatic decline".

Several typos. Some parts need to be reworded.

While a single blot is shown to confirm overexpression or knockdown of fbl, the extent of fbl OE or KD is unclear in the different tissues, but would be great to correlate with the effect size. Should be quantified.

Reviewer #3 (Remarks to the Author):

Huang et al use a drosophila model to demonstrate a new link between the PANK2 gene (the cause of PKAN neurodegenerative disease) and PINK-1 (mutations cause Parkinson's disease). Fbl (homolog of PANK2) and PINK1 loss of function mutants displayed a similar phenotype, and fbl over-expression, or introduction of pantethine (which supplies precursors to coenzyme A, which is depleted by flb) rescued the PINK1 loss of function deficits. PINK1 was shown to control the translation of fbl mRNA on the outer mitochondrial membrane. Reduced CoA/acetyl-CoA metabolism resultant from loss of function fbl or PINK1 was shown to reduced mitophagy that is regulated by ref2p/p62, which requires acetylation. Curiously, loss of function of parkin, which cooperates with PINK1 to induce mitophagy did not have the deficits in fbl, indeed they have elevated CoA, and parkin over-expression did not impact on the phenotype of loss of function fbl. I think this is a very interesting set of results, especially since it mechanistically links two related disease, PKAN and Parkinson's disease, and there are new therapeutic avenues that may emerge from these new observations. I have the following comments:

- I am convinced that PINK1 functions to promote fbl translation into the mitochondria, and that this is necessary for CoA production that is required for ref2p activity. While reduced ref2p activity will result in reduced mitophagy, it will also lead to reduced autophagy in general. Since PARKIN over-expression did not rescue the phenotype, it may suggest that reduced mitophagy was not the cause of the phenotype, rather this was caused by some other consequence of reduced autophagy, or indeed another pathway that is impacted by reduced CoA. It seems that the PINK1 and PARKIN interaction that regulates mitophagy is not important for fbl loss of function toxicity, rather PINK1's role in regulating fbl translation into the mitochondria (with reduced mitophagy just one of many consequences that result from this). The authors should demonstrate that the phenotype is ultimately due to reduced mitophagy, or another consequence of reduced ref2p activity.
- One other consequence of reduced autophagy is reduced ferritinophagy. This will result in iron elevation, which is defining feature of PKAN (Type 1 Neurodegeneration with Brain Iron Accumulation), and also Parkinson's disease (including PINK1 Parkinson's disease- e.g. PMID: 17415511). Examining the iron changes would be of interested to determine if this explains the broader phenotype of the fbl loss of function drosophila (do iron chelators ameliorate the phenotype?). It may also explain why iron is elevated in PKAN, which is currently uncertain.
- A limitation of the work is that almost all the findings are in drosophila, while the work on human cells rather limited, and only in a few panels in an extended figure. I think it is important to replicated the key findings in human cells to be sure that, for example, PINK1 has the same role in humans as it does in drosophila to regulate the translation of fbl/PANK2 into the mitochondria.

- It is interesting that pantethine supplementation improved the phenotype of *fbl* and *pink1* loss of function *Drosophila*. Another precursor of CoA that was shown to improve animal models of PKAN was fosmetpantotenate was shown not to be effective in a human clinical trial. I think the authors should discuss whether this result has implications for the likely benefit of pantethine for PKAN or PINK1-associated Parkinson's.

Reviewer #4 (Remarks to the Author):

The manuscript by Huang et al is extremely interesting and sheds new light on the molecular mechanisms involved in the pathogenesis of some forms of PD and PKAN. It describes a novel link between PINK1 activity and *fbl*/PANK2 translation at the mitochondrial membrane. Defects in PINK1 would result in lower levels of mitochondrial *fbl*/PANK2, affecting CoA production and hence the acetylation of proteins. Limited acetylation of Ref2p/p62 would negatively affect removal of damaged mitochondria.

I believe that there are a few points the authors should address in order to improve quality and strength of the manuscript.

-The authors present several immunoblotting experiments to document differences in protein levels, but they provide no quantification of the data. I suggest to add a graph for each WB experiment.

-The authors document a strong connection between PINK1 and CoA homeostasis. At the same time, silencing of enzymes involved in CoA biosynthesis (*fbl1* and *ppcdc*) has no effect on mitochondria aggregation (in muscles) and no changes on wing position and fly activity. Overexpression of *fbl* provides phenotypic rescue by increasing CoA level. To better understand the role and relevance of CoA, I suggest to compare CoA and acetyl-CoA levels in *pink1*, *fbl* and *dppcdc* silenced flies.

-Supplementation of D-pantethine rescued *dPINK1* LOF phenotypes. The authors themselves say that the action of the molecule is unclear. Pantethine has a known antioxidant activity and the rescue effect on the MPTP model of PD was associated with increased GSH production and recovery of complex I activity. Therefore, the increase in CoA level could be a secondary effect. To prove the direct connection with CoA level, the authors should feed flies with other molecules, such as 4-phosphopantetheine, known to be a direct precursor of CoA.

-The authors nicely proved that PINK1 controls the translation of *Fbl* at the OMM. At the same time *Fbl* overexpression rescues PINK1 defects. It could be interesting to verify exogenous *Fbl* localization in PINK1-silenced/mutant cells.

Minor points

-Abbreviations should be used consistently and spelled out the first time (for instance, dopaminergic neurons are abbreviated with DN or Da neurons; PPL 1cluster)

-Possible comment on the reduction of ATP content upon fbl overexpression and pantethine treatment.

-Fbl-S isoform overexpression (extended data fig 2b) appears to worsen mitochondria aggregation whereas the authors say it has no effects. Could the author correct and comment on this?

-In M&M, um is often used instead of μm

-Fig 4d; Correct PIMK1

-Results Pag 13 correct FBI and ... keep maintain mitochondria...

-The usefulness of the experiment described in the extended data figure 5f is not clear to this reviewer.

Point-by-Point Response to Reviewers' Comments

NCOMMS-20-20442-T

Reviewer's Comments:

Reviewer #1 (Remarks to the Author)

This is a very interesting paper showing a potential functional correlation between PINK1 and mitochondrial protein Pantothenate Kinase 2 in the regulation of mitochondrial quality control. The topic is relevant and can potentially interests many scientists working in this field.

We thank the high evaluation of our study from Reviewer 1.

By using fruit fly genetic as in vivo approach, the authors demonstrate a genetic interaction between mitochondrial quality control protein PINK1 and fumble (homologue of PANK2), a key enzyme in CoA metabolism. Starting from the observation that fumble mutants phenocopy PINK1, and that overexpressing fumble in PINK1 KO background rescues PINK1 KO phenotype (but not Parkin), authors conclude that the two proteins genetically interact, with PINK1 being upstream of fumble in a pathway that regulate MQC. In the proposed mechanism, Fbl regulates the acetylation of p62, thus directly promoting p62-dependent mitophagy. In addition, PINK1 controls mitochondrial anchoring and import of fbl mRNA, which is required for translation of this nuclear encoded enzyme so important for CoA synthesis. In my opinion, the fact that PINK1 affects fbl mRNA anchoring and its mitochondrial import, is per se very interesting and deserves publication. The role of fbl in the regulation of MQC is a nice addition to a very intriguing story, but as it is, the study is too preliminary for publication. I am particularly concerned about the analysis of mitophagy, the biochemical nature of Fbl-p62 interaction, and the interpretation of the epistatic studies. Please see below for details:

We thank the comments from the Reviewer. In the revised manuscript, we particularly provided strengthened evidence relating to the mitophagy regulation, Fbl/PANK2-Ref(2)P/p62 interaction, and genetic interactions to address these concerns.

(i) The epistatic analysis of PINK1-Fbl interaction is puzzling: in the double mutant (PINK1 RNAi and Fbl RNAi), authors show a significant aggravation of the phenotype which shouldn't be the case if, as proposed by the authors, Fbl and PINK1 operate in a linear pathway with PINK1 upstream of Fbl. This result actually indicates that more likely Fbl affects other pathways that are independent of PINK1 and MQC, and presumably impacts mitochondrial metabolism.

This is a good point that needs some explanation here. From genetic point of view, clear genetic epistasis relationship can only be established with NULL mutations for loss of function of genes. In this context, the worsened phenotype of double mutant (*PINK1* RNAi and *fbl* RNAi) would not constitute a strong evidence against the epistasis relationship between *PINK1* and *fbl*. On the other hand, it is not possible to analyze this with null mutants; complete absence of Fbl would not be possible because CoA is such a vital metabolic intermediate molecule without which life cannot be sustained. The observation that *PINK1* RNAi was aggravated by *fbl* RNAi could be explained in two ways: (1) they may be in parallel pathways, working synergistically or together towards a specific phenotype; (2) they may be in a linear pathway, both of which, when knocked down, become limiting. In other words, aggravation of *PINK1* RNAi by *fbl* RNAi, this fact by itself, can only indicate they are interacting, but not enough to suggest they are in a linear pathway. To establish they are in a linear pathway needs more concrete evidence. We indeed provided substantial evidence that they work linearly. For example, *fbl* overexpression (OE) can rescue

PINK1 mutant or RNAi (Fig. 2 and Extended Data Fig. 2); *PINK1* loss directly impacts *fbl* mRNA location and protein level (Fig. 7 and Extended Data Fig. 9). Considering that PINK1 loss affects Fbl localization and expression, they would not be in parallel pathways. Further evidence using *Atg* genes reinforced that the Fbl rescue of *PINK1* is through autophagy regulation.

At molecular level, the explanation for *fbl* RNAi aggravating *PINK1* RNAi is as follows: *PINK1* RNAi reduced *fbl* mRNA positioning on the mitochondria, affecting its translation and function, and on top of this, *fbl* RNAi would make this worse. This explanation fits well with the above-mentioned scenario 2, where both are in a linear pathway and when knocked down, both were limited and in combination would worsen the phenotype. Consistent with this scenario, the Fbl protein level in *PINK1* and *fbl* duo RNAi fly was dramatically lower than that of *PINK1* RNAi alone (See below, also Extended Data Fig. 9a, b). As a direct readout of Fbl activity, CoA level in *PINK1* and *fbl* duo RNAi fly was also found significantly lower than the *PINK1* RNAi fly (See below, also Fig. 2j).

Therefore, our data strongly supports that *fbl* is downstream of *PINK1*. On the other hand, Fbl, the enzyme catalyzing the key metabolic component CoA, does things more than its role in the PINK1-Fbl pathway.

(ii) Mitophagy investigation is underdeveloped. Analysis of the involvement of autophagy genes (ATG1 and Ref2p/p62) is not a read out for mitophagy. The in vivo studies proposed in fig.4 are indeed interesting and indicative of an involvement of autophagy in the mechanism of function of Fbl. Mitophagy is however a selective type of autophagy that involves the degradation of mitochondria. There are no evidence in this paper that mitochondrial degradation by autophagy (i.e. mitophagy) is differentially affected by Fbl, and in combination with PINK1. Accumulation of p62 and/or LC3/ATG8 -decorated mitochondria cannot be interpreted as an enhancement of mitophagy, as this could be the result of impaired flux (i.e. impaired degradation). Authors should clearly show increased number of mitochondria and/or mitochondrial parts undergoing degradation in this condition, paralleled by decreased mitochondrial mass (in case mitochondrial biogenesis is not affected, this should be also tested). In flies, mitochondrial degradation by the lysosomes can be specifically addressed in vivo by using mt-QC or mt-Keima previously generated for the characterization of MQC in vivo in flies (please see for example Lee JJ et al., The Journal of cell biology 2018).

We concur with the suggestive comments about mitophagy study. In the revised manuscript, we have added substantial evidence showing the involvement of mitophagy in Fbl rescue of *PINK1* mutant. Previous genetic interactions of Fbl with *Atg* genes and *ref(2)P* (Fig. 5a and Extended Data Fig. 5a; original Fig. 4a and Extended Data Fig. 5a) suggest a potential involvement of autophagy in Fbl's rescue. In the revision, we have conducted the following experiments to address the reviewer's concerns and reinforce our conclusions.

1) We examined Marf (the homolog of human Mitofusions) levels in *fbl* OE flies. As a known mitophagy marker¹, we found that Marf was significantly reduced upon Fbl induction in both *WT* and

PINK1 mutant flies (see below, also Extended Data Fig. 5b).

We also performed semiquantitative RT-PCR to analyze the transcription of mitochondrial genome encoded genes such as *mt:Col1*, *mt:Cyt-b* and *mt:ATPase6*. Two groups of flies were analyzed here. We found in *PINK1* loss-of-function (LOF) flies²; *fbL* OE dramatically restored the mitophagy activity and reduced the mitochondrial mass, while conversely, *fbL* knock-down further aggravated the phenotypes (see below, also Extended Data Fig. 5c, d). The reduction of mitochondrial mass caused by *fbL* OE in *PINK1* was mitigated by *Atg1* and *ref(2)P* knock-down, and enhanced by *Atg1* OE, indicating *fbL* rescue of *PINK1* mutant is via mitophagy (see below, also Extended Data Fig. 5e, f).

As an important control, mitochondria biogenesis was not affected, as suggested by the consistent transcription of *spargel* gene (fly homolog of *PGC-1 α*) in all genotypes (see below, also Extended Data Fig. 5g-j).

2) To visualize the mitochondrial degradation by lysosome *in vivo*, we expressed the mito-QC in the fly muscle³. Intriguingly, we found dramatically increased mitolysosomes in *PINK1* RNAi flies with *fbL* OE, indicated by the signal of “spectral shifted” puncta of more acidic conditions (mCherry) (see below, also Fig. 5d, e). This serves as direct evidence showing increased mitophagy in *fbL* OE flies.

As suggested, we quantified the mitochondrial aggregations in our study, however, the traditional tests such as ANOVA or Student's t-test were not applicable for this set of very skewed data. To adapt the skewness of our data, we described the quantifications in violin plots and used two-proportion Z test. This test is more relevant to compare the behavior of the extreme values under different experiment conditions. Examples shown below are quantifications of the signal intensity of mitoGFP aggregates in the muscle tissue of *PINK1* LOF flies with *fbl* OE and administration of D-pan (see below, also Fig. 2d, 3c), indicating elevating *fbl-L* OE or CoA synthesis in the absence of *PINK1* activity removes mitochondrial aggregates.

3) Another effort to demonstrate the importance of autophagic flux in *Fbl*'s rescue is to block it by feeding flies with chloroquine (CQ) — a specific inhibitor of autophagy. CQ administration significantly reversed the rescuing of wing posture by *fbl* OE and also resulted in accumulation of autophagy related proteins such as Ref(2)P and poly-ubiquitin in *PINK1* LOF flies (see below, also Fig. 5f and Extended Data Fig. 5k). The outcome of pharmacological inhibition of autophagy flux also supports the idea that mitophagy mediates *Fbl*'s rescue in *PINK1* LOF flies.

In conclusion, these experiments strongly support that the biological activity of *Fbl* regulates mitophagy rate in both *WT* and *PINK1* mutant flies, which is the key to mediating the phenotypic rescues of *PINK1* loss. We later also conducted experiment to verify the regulation of *Fbl* on mitophagy in human cells. Please see our responses to Reviewer 2's questions.

(iii) It is unclear how *Fbl* promotes p62 acetylation. Authors suggest a potential involvement for acetyltransferase and deacetylase Tip60 and HDAC6 but how the activity of these enzymes correlates to *Fbl* function is completely unexplored. The nature of this interaction could be direct or indirect and deserves in deep investigation.

To explore the nature of *Fbl* and Ref(2)P interaction in mitophagy, in addition to the data we presented in (Fig. 6e-g; original Fig. 4h-j), we further studied the genetic interactions between *Tip60*, *HDAC6* and *fbl* in *PINK1* LOF flies. (see below, also Extended Data Fig. 8d-f)

In *PINK1* RNAi flies, we found that *Tip60* RNAi and *HDAC6* OE enhanced wing abnormalities, while *Tip60* OE and *HDAC6* RNAi significantly mitigated them. Intriguingly, *Tip60* RNAi and *HDAC6* OE acted more potently in *fbl* OE flies (*PINK1* LOF background), while *Tip60* OE and *HDAC6* RNAi showed no further rescue, indicating that *Tip60* and *HDAC6* genes are downstream of *fbl*. On the contrary, *fbl* RNAi nearly eliminated all the actions of *Tip60* and *HDAC6* on *PINK1* RNAi flies, which can be explained by the consequence of severe reduction of acetyl-CoA, the essential element for *Tip60* and *HDAC6* functions.

In addition to the genetic interactions, we also verified the regulation of Ref(2)p acetylation by *Tip60* biochemically. We expressed and purified the Ref(2)P and *Tip60* proteins from *E.coli*, and conducted *in vitro* acetylation assay. We surprisingly found that although the K420 and K435 sites of human p62 UBA domain are not conserved, *Drosophila* Ref(2)P still can be acetylated by *Tip60*, suggesting the existence of other potential acetylation sites that can be recognized. (see below, also Extended Data Fig. 8i). Our attempt of deacetylation assay failed due to the insoluble inclusion bodies in *HDAC6 E.coli* expression.

All together, we conclude that *Tip60* and *HDAC6* regulate the acetylation of Ref(2)P and subsequently affect the initiation of mitophagy in *Drosophila*.

Reviewer #2 (Remarks to the Author)

While rare, Pantothenate Kinase-Associated Neurodegeneration is the most common form of neurodegeneration with brain iron accumulation (NBIA) and is caused by loss of PANK2 function. PANK2 (or *fumble/fbl* in flies) encodes a mitochondrial form of pantothenate kinases, which catalyze the first/key step of CoA synthesis. Here the authors quite convincingly show a genetic interaction between PANK2 and PINK1 homologs in flies. PINK1 and Parkin together mediate a conserved neuroprotective mitochondrial autophagy pathway, but are both mutated and lost in early-onset Parkinson's disease. *Fbl* and *dPINK1* mutants/RNAi display similar morphologic and functional mitochondrial abnormalities (at least in neurons), and overexpression of the mitochondrial, but not cytosolic (or kinase dead) *Fbl* isoform partially restored phenotypes of PINK1 loss of function, whereas *Fbl* RNAi exacerbates these. Likewise, *d*-pantethine supplementation restores CoA levels and is able to partially rescue PINK1 phenotypes as well. This is a quite solid part of the study, but the mechanistic aspects are very limited and less convincing and will need substantial work to be backed up. This includes both additional biochemical and immunofluorescence validation and adequate quantification of the respective effects. Moreover, it will be essential to clearly demonstrate conservation of the effects and mechanisms beyond *Drosophila*, especially given that the authors try to link this to human disease.

We appreciate the positive comments on our study from Reviewer 2. In the revised manuscript, we conducted more experiments to study the mechanism of how *Fbl* activity and CoA metabolism regulate mitophagy and rescue the defects of *PINK1* mutant. We quantified essential immunoblots and immunofluorescence staining to illustrate the indicated changes such as Fig. 1b, 2d, 2f, 3d, 3e, 4d, 4f, 5e, 5h, 5j, 5l, 7b, 7f, 7i, 7k, 7m and Extended Data Fig. 1b, 1d, 2c, 3o, 5d, 5f, 5h, 5j, 6b, 6d, 6f, 6h, 6j, 6l, 6p, 7b, 7d, 7f, 7h, 8h, 9b, 9c, 9g, 9j, 9n, 9p.

In order to test the conservation of the mechanism of *Fbl*/PANK2 and CoA assisting mitophagy, we checked whether the rescue of *PINK1* loss in the flies could be reproduced in human cells. We performed 1) immunofluorescent staining of LC3, Lamp2 and p62, which showed an increase of mitophagy by PANK2 OE and *D*-pan treatment; 2) mito-Keima live imaging, which showed a facilitation of mitophagy by PANK2 and *D*-pan in *PINK1* knock-down cells; 3) immunoblotting, which showed a rescue of mitophagy defect upon CCCP treatments.

1) LC3, Lamp2 and p62 staining (see below, also Fig. 5g-j and Extended Data Fig. 6a-d, g-j)

2) mito-Keima live imaging (see below, also Extended Data Fig. 6n)

3) Immunoblots (see below, also Extended Data Fig. 6e, f, k, l)

Conversely, *PANK2* knock down significantly impaired the execution of mitophagy. Supporting evidence comes from both immunofluorescence staining of LC3, Lamp2 and p62, and immunoblotting of mitochondrial and autophagy markers.

1) LC3, Lamp2 and p62 staining (see below, also Fig. 5k, l and Extended Data Fig. 7a-d)

2) Immunoblot (see below, also Extended Data Fig. 7e-f)

We have performed additional experiments in *Drosophila* models to investigate how mitophagy is mechanistically involved in rescue of PD models with Fbl/*PANK2* and CoA supplement. This part of our new data can be found in our response to Reviewer 1, point (ii).

Further it is quite puzzling that *fbl*/PANK2 seem to operate downstream of PINK1, but in parallel to Parkin, although it is then implicated in pathways that are regulated by both, PINK1 and Parkin, such as mitophagy or the localized mitochondrial translation of select mRNAs. This needs to be substantiated too. We thank Reviewer 2's comments on this point. In the revised manuscript, we have further analyzed the epistatic interaction between *PINK1* and *fbl*, please see our response to Reviewer 1 point (i). The interaction between *parkin* and *fbl* is complicated. We hypothesized that the diverse rescue of *PINK1* and *parkin* mutants by *fbl* OE is due to the differential effects of PINK1 and Parkin on CoA and acetyl-CoA levels (Fig. 2h, i and 4i, j; original Fig. 2f, g and Extended Data Fig. 4f, g). Unlike *PINK1* mutant (both CoA and acetyl-CoA levels are decreased), the CoA level of *parkin* mutant is actually increased (only acetyl-CoA level is decreased), in line with the observation of loss of PDH activity in human cell *parkin* model⁴. Therefore, elevating CoA production in *parkin* mutant by *fbl*/PANK2 OE would not further promote mitophagy in this scenario. In summary, Parkin has possibly an additional role in acetyl-CoA control, which is downstream of CoA and beyond the function of PINK1 and Fbl. Further information on this can be found in our responses to Question 3 and 7 below.

Overall this is an interesting manuscript with novel findings and potential disease relevance, but I have several major concerns that need to be addressed:

1) mitochondrial aggregation phenotype (labeled by mitoGFP)

This phenotype is actually never quantified, so here we rely on single immunofluorescence images for both TH neurons and muscles. A more objective measure such as of number/size of the aggregates or GFP intensity is needed to back up claims that these aggregates are “prominent” or “correlate with...”.

Please also see our response to Reviewer 1 point (ii). To address this concern, we quantified the size/area of aggregates in both *Drosophila* DA neurons and indirect muscle tissues. Intriguingly, we found that the ANOVA test and Student's t-test are not suitable for such very skewed data. To adapt to the skewness of our data, we described our data in violin plots and used two-proportion Z test. More information can be found in the DISCUSSION section. For example, data shown below are quantifications of the signal intensity of mitoGFP aggregates in the muscle tissue of *PINK1* LOF flies with *fbl* OE and administration of D-pan (see below, also Fig. 2d, 3c), indicating elevating *fbl*-L OE and D-pan supplement in *PINK1* mutant removes mitochondrial aggregates. Quantification data (Fig. 1b, 2d, 2f, 3d, 3f, 4d, 4f and Extended Data Fig. 2c, 3o) supports our statements and hypothesis in this paper.

Also Fig 1a clearly shows similar aggregates for the Fbl-L OE strain although it is stated: ‘Prominent mitochondrial aggregations (labeled by mitoGFP) were observed in the DA neurons of Fbl RNAi flies, reminiscent of that in dPINK1 mutant (Fig. 1a), while Fbl-L OE did not generate any notable abnormality (Fig. 1a).’

We thank the reviewer's comments here. We agree that mild/small mitochondrial aggregates can be occasionally found in DA neurons of normal flies. However, they are rare and transient. In our hands, *fbl*-L OE only caused very mild alteration. To avoid the confusion, we repeated this experiment and replaced *WT* and *fbl*-L OE group with new and better representing images (see below, also new Fig. 1a).

2) Complex I

How do the authors explain a reduced complex I dependent respiration, but no reduction of complex I activity? Please explain including the differences between *PINK1* and *fbl* phenotypes and rescue regarding complex I.

The rate of complex-I dependent respiration is determined by the activity of all sub-complexes in RCC (respiratory Chain Complex), including complex-I, -III, and -IV. Similarly, complex-II dependent respiration requires complex-II, -III and -IV, while complex-III dependent respiration needs complex-III and -IV. In our study, we found the reduced complex-I dependent respiration (Fig. 1e), but no reduction of complex-I activity (Fig. 1h).

This observation suggests that the defect of RCC caused by *fbl* LOF should come from complexes other than complex-I, that is, the complex-III or -IV. Consistent with this, complex-II and complex-III dependent respiration rates (both require complexes-III and -IV) were found to be reduced (Fig. 1f, g). These imply complex-I is insensitive to Fbl activity or cellular CoA levels. In *PINK1* mutant, by contrast, defects in RCC were more general, and a compromised complex-I activity was observed (Extended Data Fig. 2h-l), likely as a result of *PINK1*'s broader role in affecting mitochondria functions (besides regulating *Fbl/Pank2* as shown later).

In support of this notion, *fbl-L* OE rescued complex-II dependent respiration (-II, -III, -IV) and complex-I dependent respiration (-I, -III, -IV), while it didn't rescue complex-I activity at all (Extended Data Fig. 2h-l; original Extended Data Fig. 2e-h). To further confirm this hypothesis, in the revision, we also found that yeast *NDI1* (NADH:ubiquinone oxidoreductase, a counterpart of higher eukaryotic multi-subunit respiratory complex-I) could not rescue *fbl* RNAi fly (see below, also Extended Data Fig. 1h); in contrast to its nice rescue of *PINK1* mutant in our previous study⁵.

3) Parkin

Despite the clear genetic interaction between *PANK2* and *PINK1*, lack of the same for *Parkin* is very puzzling and needs further experimental validation and additional discussion. Although it appears that neither *Fbl* overexpression, nor d-pantethine supplementation rescued parkin KO phenotypes, several questions remain some of which are:

Please see our point-to-point response to individual questions below. In general, the differential roles of *PINK1* and *Parkin* in regulating the level of cellular CoA and its derivants acetyl-CoA is the key to

understanding the puzzle. Compared to *PINK1* mutant, *Drosophila parkin* mutant has a quite different CoA metabolism. Unlike the *PINK1* mutant (Fig. 2h, i; original Fig. 2f, g), *fbl* OE could not change the CoA and acetyl-CoA levels in *parkin* mutant (Fig. 4i, j; original Extended Data Fig. 4g, h), which implies a possible blockage exists in the conversion of acetyl-CoA from CoA when Parkin is affected.

Does *fbl* RNAi exacerbates Parkin KO phenotypes, similar to a *PINK1* mutant background?

To answer this question, we ubiquitously knocked down *fbl* in *Parkin* mutant and found strong synergistic effects between *fbl* and *parkin*. Double knock out of *fbl* and *parkin* resulted in pupal semi-lethality, while neither single mutant died this early (see below, also Extended Data Fig. 9q).

Does Parkin OE restore the acetyl/CoA levels in *PINK1* KO animals?

We included new data in the revised manuscript to answer this question. We found that *parkin* OE restored the CoA and acetyl-CoA levels in *PINK1* mutant flies, whereas it did not elevate CoA or acetyl-CoA level in *WT* flies. (see below, also Extended Data Fig. 4d, e).

What are the acetyl/CoA levels in *PINK1*/*Parkin* double mutants?

The double mutant of *PINK1/Y; parkin^{1/Δ21}* was lethal in our pilot test. Instead, we measured the CoA and acetyl-CoA levels in *PINK1, parkin* double knock-down (RNAi) flies, as shown above (Extended Data Fig. 4d, e). Although double knock-down mutant was unhealthy and exhibited severer acetyl-CoA deficit, its CoA level was higher than *PINK1* RNAi fly, which was actually reminiscent of *parkin* mutant (see data above). The underlying reason may be related to the strong loss of PDH activity under Parkin deficiency condition⁴. We discussed it in detail in our response to question 7 below.

Also this investigation should be expanded to manipulation p62 and Atg genes in *Parkin* KOs

We appreciate this suggestion. The relationships of *ref(2)P* and *Atg* genes such as *ATG1* with *parkin* have been extensively studied since all of them are considered as important players in mitophagy. *Atg1* OE can effectively suppress tissue degeneration in *PINK1/parkin* mutants by promoting mitochondrial fission in *Drosophila*⁶. *Ref(2)P* has also been shown to be required for Parkin-mediated rescue of mitochondrial defects in *PINK1* mutant⁷. Since *fbl* OE cannot rescue the defects of *parkin* mutant, likely due to Parkin's additional involvement in CoA conversion to acetyl CoA, we did not pursue much further the effects of *fbl* OE on these genes in *parkin* mutant.

Are PANK2 levels affected in Parkin mutant flies/cells?

Yes. Like PINK1, Parkin also significantly regulates Fbl metabolism in *Drosophila*. In *parkin* mutant, both the recruitment of mRNA on mitochondria and protein translation of *fbl* are reduced. (see below, also Extended Data Fig. 9m-o).

4) decline in mitophagy:

The authors actually never measure mitophagy, but rely on western blot readouts or single immunofluorescent images. Neither of these can inform on mitophagy which is a dynamic process and must be measured as such. Also none of that data shown for p62, ubiquitin or LC3 has been quantified, but all findings need to be corroborated both biochemically and by immunofluorescence to corroborate the findings in flies and human cells. How do authors explain the lack of genetic interactions with parkin in the context of mitophagy?

We appreciate this criticism from Reviewer 2. In the revised manuscript, we provided new data to support the idea that mitophagy actively involves in the rescue of *fbl* OE in *PINK1* mutant flies including qualifications of autophagy markers such as mito-QC. Please see our response to Review 1, point (ii). We also conducted new experiments in human cells to verify the conservation of mechanism. Please see our previous response.

Although Parkin is considered as an executor in the PINK1-Parkin mediated mitophagy pathway, its main function is to ubiquitinate targets on the outer membrane of mitochondria. This is a downstream step of PINK1 activation, independent of autophagy receptor (Ref(2)P and p62) activation, which is regulated by Fbl activity and bioavailable CoA/acetyl-CoA levels. In *PINK1* mutant, *fbl* OE or D-pan administration rescues the mitophagy defect from the elevated cellular CoA and acetyl-CoA levels. The restored acetyl-CoA level activates autophagy receptor Ref(2)P/p62 and facilitates its binding to the ubiquitinated proteins (provided by the retained Parkin) on mitochondrial surface. Whereas in *parkin* mutant, CoA has been already accumulated while acetyl-CoA level is reduced, so neither *fbl* OE nor D-pan treatment can provide further benefits to disease models in this regard. As direct evidence to support this, we compared the CoA and acetyl-CoA levels of *parkin* mutant with or without *fbl* OE. We found that CoA and acetyl-CoA levels in *parkin* flies with *fbl* OE remained the same (Fig. 4i, j, original Extended Data Fig. 4g, h). In addition, loss-of-function of Parkin completely abolishes the ubiquitination of mitochondrial outer membrane proteins, which leaves no anchor for Ref(2)P/p62 proteins, even they are activated. In short, *fbl* OE and D-pan administration will not rescue the *parkin* mutant. In *fbl* mutant, defects from a global metabolic dyshomeostasis due to CoA deficiency and accumulation of cysteine-containing substrates are expected, which may not be effectively reversed by just regulating mitophagy. Along this line, we do not expect *parkin* OE to rescue *fbl* mutant or CoA synthesis defect from loss of Fbl activity.

5) human cells.

Fibroblasts are only N=1 for each mutant and controls. Data needs to be shown on the same western blot and needs quantification for each genotype and treatment. If there are no additional samples, this at least needs technical repeats and a more thorough investigation over time.

In the revised manuscript, three PKAN fibroblasts are compared with control fibroblasts (see below, also Extended Data Fig. 7g, h). The original image in Fig. S5D was replaced with the new data.

Anything with PINK1 or knockdown in human cells needs to be controlled by +/- CCCP treatment and blotting for endogenous PINK1, Parkin, and phosphorylated ubiquitin which are all determinant of mitophagy. Also PANK2 protein levels need to be shown.

To address this concern, we used more markers in the new blots, including PANK2 and PINK1, to support the involvement of mitophagy in the process. Please see our responses to the question about mitophagy above. Some antibodies did not work well in our hands. However, when combing data from both immunoblotting and immunofluorescence staining, we are confident in the conclusion.

Ext Fig 6C: There really shouldn't be any endogenous full-length PINK1 protein detectable in unstressed human cells (input).

We understand Reviewer 2's concern about the PINK1 signal. As a kinase anchored on the outer membrane of mitochondria, PINK1 is thought to be quickly processed by mitochondrial proteases (e.g. PARL) and degraded by proteasome on healthy mitochondria. Therefore, endogenous full-length PINK1 is almost undetectable. However, depending on the antibody and experimental conditions, sometimes a weak full-length band of PINK1 can still be seen; for instance the paper published by Przedborski group on *PNAS*², and also in our previous publication (Fig. 5E)⁸. Therefore, we still believe that the PINK1 signal in Extended Data Fig. 9h (original Extended Data Fig. 6c) is real PINK1. In addition, the PINK1 signals recognized by our antibody can be specifically decreased by *PINK1* shRNA knockdown (please see our data and response to mitophagy analyses above).

To further address this concern, in the revised manuscript, we ectopically expressed HA-tagged human PINK1 in HEK cells and verified the direct binding between PINK1 protein and *PANK2* mRNA (see below, also Extended Data Fig. 9i, j). Moreover, we updated data in Extended Data Fig. 9k (original Extended Data Fig. 6d). The results indicate that full-length hPINK1 can preferentially interact with *fb/l* mRNA in mRNA-ribonucleoprotein (mRNP) complex.

6) p62 acetylation:

It is highly likely that there is an effect on overall protein acetylation and that this is not exclusively specific to p62. This line of research needs to be substantiated by other means and must be expanded to include changes in total cellular acetyl-lysine levels as well as other known controls such as tubulin and histones. In order to confirm a specific effect of acetyl-p62, for instance acetylation-mimic and -deficient version of p62 can be studied.

This is an important question. To analyze global changes in protein acetylation, we performed immunoblotting with a pan anti-acetyl lysine antibody (ab80178). Interestingly, overall cellular acetylation on protein lysine residues did not significantly change when *fb1* expression was regulated in WT flies, while specific alterations could occur on individual bands, suggesting Fbl modulation may exert differential effect on the acetylation of different targets. (see below, also Extended Data Fig. 8c).

As for testing p62 acetylation-mimic (K to Q) and -deficient (K to R) mutants, it would be better to do these *in vivo*. We first tested these p62 mutants in human cells. The p62 K2R mutant lost the ability to relocate to damaged mitochondria (see blow, also Extended Data Fig. 8g, h) indicating a similar mechanism to other autophagy processes⁹. However, the K420 and K435 sites in human p62 UBA domain are not conserved in *Drosophila* (please see the figure below). Despite the lack of conservation on these sites, we still can see biochemical acetylation on Ref(2)P by Tip60 (please see our response to Reviewer 1, point iii), which suggests the existence of potential

Lysine (K) residues are highlighted in yellow.	
human : 391	DPRLIESLSQMLSMGFSDGEGWLT ^{RL} LQT ^K NYDIGAALDTIQYS ^K H 436
	D + +S+ M++MGFS+EG WLT+LL++ +I AALD + S++
Drosophila : 552	DESIN ^K SIHAMMAMGFSNEGAWLTQLLESVQGNISAAALDVMNVSQN 597

modifications on the other (nearby) lysine sites through a similar mechanism. The exact mechanism can be explored in the future.

7) RNA:

The mRNA levels in different fractions needs to be shown on the same blot/gels alongside proper positive and negative controls and need quantification by qPCR. Likewise, CLIP needs to be quantified when comparing PINK1 WT and mutant, relative to the amount of IP'ed PINK1 protein.

Thank for Reviewer 2's suggestion on this point. Although the main purpose here is to compare them individually with their controls, we agree that showing them on the same blot with quantification would provide more compelling support to our hypothesis. Per this suggestion, in the revised manuscript, we replaced the old figures with new data that fits the criteria. (see the figures below, also Fig. 7e-m and

Extended Data Fig. 9f-k; original Fig. 5b-f and Extended Data Fig. 6b-d).

Also in this context, how do authors explain their Parkin findings as both PINK1 and Parkin were suggested to coordinately derepress translation of OXPHOS-related mRNAs by targeting the Glo for ubiquitination.

This question is related to question 3. Please see our response above. Although both PINK1 and Parkin regulate *fbl* translation, their roles in regulating CoA and acetyl-CoA are quite different (please see our response at the beginning and to Question 3). The *parkin* mutant has a higher CoA level, thus it is insensitive to further Fbl upregulation. It has been reported that *PRKN* knockdown in human H460 cells impairs the activity of PDH (pyruvate dehydrogenase) complex and results in acetyl-CoA level drop⁴. We suspect that the dysfunction of pyruvate dehydrogenase is also responsible for the CoA accumulation and acetyl-CoA decline in fly *parkin* mutant. To verify this, we knocked down and overexpressed *Pdha* in *parkin* LOF flies via the GeneSwitch GAL4 system (because *Pdha* knockdown severely influenced fly development), and compared their CoA and acetyl-CoA levels. Intriguingly, *Pdha* knockdown strongly enhanced CoA accumulation, while *Pdha* overexpression mildly mitigated it; whereas *Pdha* knockdown further reduced acetyl-CoA level and *Pdha* overexpression rescued it when *parkin* was suppressed (see below, also Extended Data Fig. 4a, b). Our interpretation is that *Pdha* knockdown further decreases its activity to disturb the conversion of CoA to acetyl-CoA, and the effect of Parkin on *Pdha* appears to be through regulation of the enzymatic activity rather than translation. (see below, also Extended Data Fig. 4c)

Minor:

“Dramatic decline of complex I-, II- and III- dependent respiration was observed in Fbl LOF flies (Fig. 1d-1f), also similar to that in the dPINK1 LOF fly.” This looks like an approximately 25-50% reduction, so mild to moderate decrease would perhaps be more appropriate than “dramatic decline”.

Several typos. Some parts need to be reworded.

We thank the reviewer for pointing out these minor issues and typos here. They have been corrected in the revised manuscript. We also changed the word “dramatic decline” to “moderate decrease”.

While a single blot is shown to confirm overexpression or knockdown of *fbl*, the extent of *fbl* OE or KD is unclear in the different tissues but would be great to correlate with the effect size. Should be quantified. Per suggestion, we performed RT-PCR to analyze the efficacy of *fbl* OE and RNAi in whole fly (*Da-Gal4*), muscle (*MHC-Gal4*), and brain (*Elav-Gal4*). Please see our data below (also Extended Data Fig. 1c, d).

Reviewer #3 (Remarks to the Author)

Huang et al use a drosophila model to demonstrate a new link between the PANK2 gene (the cause of PKAN neurodegenerative disease) and PINK-1 (mutations cause Parkinson's disease). Fbl (homolog of PANK2) and PINK1 loss of function mutants displayed a similar phenotype, and fbl over-expression, or introduction of pantethine (which supplies precursors to coenzyme A, which is depleted by *fbl*) rescued the PINK1 loss of function deficits. PINK1 was shown to control the translation of fbl mRNA on the outer mitochondrial membrane. Reduced CoA/acetyl-CoA metabolism resultant from loss of function fbl or PINK1 was shown to reduced mitophagy that is regulated by ref2p/p62, which requires acetylation. Curiously, loss of function of parkin, which cooperates with PINK1 to induce mitophagy did not have the deficits in fbl, indeed they have elevated CoA, and parkin over-expression did not impact on the phenotype of loss of function fbl. I think this is a very interesting set of results, especially since it mechanistically links two related disease, PKAN and Parkinson's disease, and there are new therapeutic avenues that may emerge from these new observations.

We thank the high evaluation from Reviewer 3.

I have the following comments:

- I am convinced that PINK1 functions to promote fbl translation into the mitochondria, and that this is necessary for CoA production that is required for ref2p activity. While reduced ref2p activity will result in reduced mitophagy, it will also lead to reduced autophagy in general. Since PARKIN over-expression did not rescue the phenotype, it may suggest that reduced mitophagy was not the cause of the phenotype, rather this was caused by some other consequence of reduced autophagy, or indeed another pathway that is impacted by reduced CoA.

It seems that the PINK1 and PARKIN interaction that regulates mitophagy is not important for fbl loss of function toxicity, rather PINK1's role in regulating fbl translation into the mitochondria (with reduced mitophagy just one of many consequences that result from this). The authors should demonstrate that the phenotype is ultimately due to reduced mitophagy, or another consequence of reduced ref2p activity.

We thank Reviewer 3 for bringing up this possibility. At this point, we agree with the reviewer that declined autophagy/mitophagy or Ref(2)P activity may not be the main source of toxicities/defects of *fbl* mutant. The point raised in our study is that in *PINK1* mutant (PD model) reduced *fbl* mRNA translation causes CoA and acetyl-CoA deficiency and leads to mitophagy obstruction. The translational regulation of *PANK2* by PINK1 may be the key to alleviating mitochondrial defects caused by insufficient Pank2 expression in Parkinson's disease. Since PANK2/Fbl is such an essential kinase for *de novo* CoA synthesis, its dysfunction will inevitably interfere with many key pathways. Therefore, the defects in *fbl* mutant may come from a combination of multiple metabolic disturbance. However, one thing we want to emphasize is that a subset of mitochondrial dysfunction of *fbl* mutant is due to reduced mitophagy. In our revision, we conducted the experiments of combining *ref(2)P* and *Atg* genes with *fbl* RNAi in flies. We found that none of them could effectively rescue the developmental defects caused by *fbl* LOF. Please see the data below (also Extended Data Fig. 5m). On the other hand, Fbl rescue of PINK1 loss appears through mitophagy regulation as ATG loss would eliminate this rescue. Please see our data in Fig. 5 and Extended Data Fig. 5, 6.

• One other consequence of reduced autophagy is reduced ferritinophagy. This will result in iron elevation, which is defining feature of PKAN (Type 1 Neurodegeneration with Brain Iron Accumulation), and also Parkinson’s disease (including PINK1 Parkinson’s disease- e.g. PMID: 17415511). Examining the iron changes would be of interested to determine if this explains the broader phenotype of the *fbl* loss of function drosophila (do iron chelators ameliorate the phenotype?). It may also explain why iron is elevated in PKAN, which is currently uncertain.

Previous studies have shown iron accumulation and alteration of Ferritin metabolism in *fbl* mutant¹⁰. In

	Mitochondrial iron content	
Da-Gal4>GFP w/ DMSO	1.05625	ng/ug protein
Da-Gal4>GFP w/ Rapamycin	0.752381	ng/ug protein
Da-Gal4>Fbl RNAi w/ DMSO	1.671924	ng/ug protein
Da-Gal4>Fbl RNAi w/ Rapamycin	0.836538	ng/ug protein

our revision, to further explore this possibility, we purified mitochondria from *fbl* mutants and subsequently measured their iron contents by Inductively Coupled Plasma Mass Spectrometry (ICP-MS). We found that iron accumulates in the mitochondria of *fbl* RNAi flies, and activation of the autophagy pathway by rapamycin treatment significantly reduces it (please see our data below).

Unfortunately, due to the low mitochondria yield of *fbl* RNAi flies, we were unable to collect enough samples for triplicate assays. We additionally compared the mRNA level of mitochondrial ferritin (mitoferritin, *Fer3HCH*), which is tightly controlled by mitochondrial iron concentration^{11,12}. Consistent with iron measurement, *Fer3HCH* mRNA levels

in *fbl* RNAi and *PINK1* RNAi flies were significantly increased, and rapamycin treatment mitigated them (please see our data below).

In *in vivo* experiments, iron chelator DFO exhibited very slight benefits to *fbl* mutant¹³. During our revision process, we also tested multiple iron chelators in *PINK1* mutants. Deferoxamine (DFO) and Bathophenanthrolinedisulfonic acid (BPS) robustly rescued the abnormal wing failure caused by *PINK1* LOF (Data not shown). This, together with our previous observation that *Fer3HCH* RNAi significantly rescued mitochondrial defects in the *PINK1* LOF *Drosophila*¹⁴, suggest iron regulation may be effective in suppressing some of *PINK1* and *fbl* LOF defects.

In sum, *fb1* mutant does present iron homeostasis abnormality, but rescuing it with iron chelation strategy may offer at most slight benefit. This appears reasonable when considering that the fundamental defect of *fb1* is CoA synthesis, a defect with many consequences. Our data did support the idea that iron homeostasis contributes to the pathogenesis of *PINK1* PD model, but substantial work is still required to make a strong argument. Because studying iron homeostasis in these diseases is beyond the scope of our current research plan, we decided to provide the relevant data as mentioned above only in our response letter but not in the revised manuscript.

- A limitation of the work is that almost all the findings are in *Drosophila*, while the work on human cells rather limited, and only in a few panels in an extended figure. I think it is important to replicate the key findings in human cells to be sure that, for example, *PINK1* has the same role in humans as it does in *Drosophila* to regulate the translation of *fb1*/PANK2 into the mitochondria.

We agree extending some of the key findings to human cells would make the story more interesting. In the revised manuscript, we provide substantial data to validate some of our findings in human cells. Please see our previous responses to Reviewer 1 and 2. The new data of human cells are in Fig. 5 and Extended Data Fig. 6-9.

- It is interesting that pantethine supplementation improved the phenotype of *fb1* and *pink1* loss of function *Drosophila*. Another precursor of CoA that was shown to improve animal models of PKAN was fosmetpantotenate was shown not to be effective in a human clinical trial. I think the authors should discuss whether this result has implications for the likely benefit of pantethine for PKAN or *PINK1*-associated Parkinson's.

We thank Reviewer 3 for her/his suggestion on this point. Since fosmetpantotenate (RE-024) is not commercially available, a small amount of RE-024 was synthesized in the lab. We tested its efficacy in *PINK1* LOF flies and human cells. We found that supplementation with RE-024 significantly promoted mitophagy in *PINK1* knockdown human cells, as indicated by the colocalization between LC3 and mitochondrial marker. In *Drosophila*, RE-024 treatment moderately but significantly rescued the abnormal wing posture and mitochondrial aggregations in *PINK1* loss-of-function flies. (see below, also Extended Data Fig. 3m-o, 6o-p)

Altogether, our results indicate that fosmetpantotenate (RE-024) has the ability to promote mitophagy and rescue mitochondria defects in *PINK1*-related PD models similar to D-Pan. In a recent clinical trial, FOSmetpantotenate Replacement Therapy (FORT) was shown not to be very effective¹⁵. In another single-arm, open-label study, pantethine was shown to delay the progression of motor dysfunction in children with PKAN, but not rescue¹⁶. The pathways of *de novo* synthesis, transport and metabolism of CoA, as well as relevant precursors are still not fully understood, and the differences between animal models (*Drosophila* and mouse) and humans may cause divergent efficacy of disease treatment. It will be interesting to further confirm their benefits in other systems such as PKAN or PD patients' fibroblast-derived neurons. We added these discussions in our revised manuscript.

Reviewer #4 (Remarks to the Author):

The manuscript by Huang et al is extremely interesting and sheds new light on the molecular mechanisms involved in the pathogenesis of some forms of PD and PKAN. It describes a novel link between PINK1 activity and *fbl*/PANK2 translation at the mitochondrial membrane. Defects in PINK1 would result in lower levels of mitochondrial *fbl*/PANK2, affecting CoA production and hence the acetylation of proteins. Limited acetylation of Ref2p/p62 would negatively affect removal of damaged mitochondria. I believe that there are a few points the authors should address in order to improve quality and strength of the manuscript.

We thank Reviewer 4 for the high evaluation of our work. We completed the recommended experiments in an effort to improve our manuscript.

-The authors present several immunoblotting experiment to document differences in protein levels, but they provide no quantification of the data. I suggest to add a graph for each WB experiment.

We added quantifications for key immunoblots, RT-PCRs and images of immunofluorescence staining of *Drosophila* tissues and human cells in our revised manuscript. For details, please also see our response to Reviewer 2.

-The authors document a strong connection between PINK1 and CoA homeostasis. At the same time, silencing of enzymes involved in CoA biosynthesis (*fbl* and *ppcdc*) has no effect on mitochondria aggregation (in muscles) and no changes on wing position and fly activity. Overexpression of *fbl* provide phenotypic rescue by increasing CoA level. To better understand the role and relevance of CoA, I suggest to compare CoA and acetyl-CoA levels in *pink1*, *fbl* and *dppcdc* silenced flies.

In our previous study, we found *fbl* RNAi has no effect on fly wing posture, jumping/flight ability and mitochondria morphology (or aggregation) in muscle tissue. *Ppcdc* RNAi enhanced the wing posture defect of *PINK1* RNAi flies, and *fbl* OE could not reverse it, suggesting the importance of CoA *de novo* synthesis in the pathogenesis of PINK1-related PD. To further evaluate relevance of CoA in this regulation as suggested, we compared the CoA and acetyl-CoA levels of *PINK1*, *fbl* and *Ppcdc* LOF flies. Since *Ppcdc* RNAi causes pupal lethality, we measured the CoA and acetyl-CoA contents in 3rd instar larvae. Consistent with our conjecture, LOF of all three genes leads to reduction of CoA and acetyl-CoA, with *Ppcdc* RNAi being the strongest. (see below, also Extended Data Fig. 2f, g)

-Supplementation of D-pantethine rescued dPINK1 LOF phenotypes. The authors themselves say that the action of the molecule is unclear. Pantethine has a known antioxidant activity and the rescue effect on the MPTP model of PD was associated with increased GSH production and recovery of complex I activity. Therefore, the increase in CoA level could be a secondary effect. To prove the direct connection with CoA level, the authors should fed flies with other molecules, such as 4-phosphopantetheine, known to be direct precursor of CoA.

To address this question, we fed PD models with other precursors of CoA in its *de novo* synthesis pathway,

fosmetpantotenate (RE-024) and 4'-phosphopantetheine (4'-PPT). Please refer our response to Reviewer 3 question 4 for data on fosmetpantotenate. Similar to fosmetpantotenate, 4'-phosphopantetheine also significantly rescued the abnormal wing posture and extended lifespan in *PINK1* mutant, but not in *parkin* mutant. Similar effects of D-pantethine, fosmetpantotenate and 4'-phosphopantetheine strongly indicate the rescue is via the CoA synthesis pathway. (see below, also Extended Data Fig. 3k, l and 4f, g)

-The authors nicely proved that PINK1 controls the translation of Fbl at the OMM. At the same time Fbl overexpression rescues PINK1 defects. It could be interesting to verify exogenous Fbl localization in PINK1-silenced/mutant cells.

We suspect that Reviewer wants to check the Fbl localization in *PINK1* silenced flies. Thus, we conducted the endogenous antibody staining in fly muscle tissue. Surprisingly, we found that although the total amount of Fbl protein is reduced, Fbl still colocalizes with mitochondria, especially aggregated mitochondria in *PINK1* RNAi flies, suggesting that the aggregated mitochondria may be more functionally active and with high membrane potential. Notably, these mitochondria showed a high membrane potential TMRM staining indicating the dependence of *fbl* mRNA translation on mitochondrial potential. And our observation raised a question that is the aggregated mitochondrial with high mitoGFP signal are actually formed in a complimentary protective way. (see below, also Fig. 7c and Extended Data Fig. 9e)

Minor points

-Abbreviations should be used consistently and spelled out the first time (for instance, dopaminergic neurons are abbreviated with DN or Da neurons; PPL 1 cluster)

We thank the suggestion from the Reviewer. We have corrected these items in the revised manuscript.

-Possible comment on the reduction of ATP content upon *fbl* overexpression and pantethine treatment. We think it may come from the excess mitophagy induced by *fbl* OE and D-pan administration, which is similar to the phenotype in *Atg1* OE flies (our unpublished data). As a proof, we detected stronger

mitophagy in *fbl* OE flies (please see our response to Reviewer 1, point ii).

-Fbl-S isoform overexpression (extended data fig 2b) appears to worsen mitochondria aggregation whereas the authors say it has no effects. Could the author correct and comment on this?

Overexpression of Fbl-S isoform cannot rescue mitochondrial aggregation in *PINK1* RNAi flies. In the revised manuscript, we replaced the original images with a low-magnification photo to better reflect the effects of Fbl-S isoforms on muscle mitochondria and also provided the quantification. (see below, also Extended Data Fig. 2b, c)

-In M&M, um is often used instead of µm

We have proofread our new manuscript to correct typos or mis-formatting.

-Fig 4d; Correct PIMK1

We corrected it in the new manuscript.

-Results Pag 13 correct FBI and ... keep maintain mitochondria...

Corrected. Thank you for pointing them out.

-The usefulness of the experiment described in the extended data figure 5f is not clear to this reviewer. The original data in the extended data figure 5f shows that benefits of D-pan also apply to human cells (SK-N-BE(2)-M17) with high levels of tyrosine hydroxylase activity and moderate dopamine-b-hydroxylase activity. We agree with the reviewer that it may not intimately connect with our main story. We deleted it in our revised manuscript.

REFERENCES:

- 1 Chen, Y. & Dorn, G. W., 2nd. PINK1-phosphorylated mitofusin 2 is a Parkin receptor for culling damaged mitochondria. *Science* **340**, 471-475, doi:10.1126/science.1231031 (2013).
- 2 Cristofol, V.-B. *et al.* PINK1-dependent recruitment of Parkin to mitochondria in mitophagy. doi:10.1073/pnas.0911187107 (2010).
- 3 Lee, J. J. *et al.* Basal mitophagy is widespread in *Drosophila* but minimally affected by loss of Pink1 or parkin. *The Journal of cell biology* **217**, doi:10.1083/jcb.201801044 (2018).
- 4 Zhang, C. *et al.* Parkin, a p53 target gene, mediates the role of p53 in glucose metabolism and the Warburg effect. *Proceedings of the National Academy of Sciences of the United States of America* **108**, doi:10.1073/pnas.1113884108 (2011).
- 5 Wu, Z. *et al.* Tricornered/NDR Kinase Signaling Mediates PINK1-directed Mitochondrial Quality Control and Tissue Maintenance. *Genes & development* **27**, doi:10.1101/gad.203406.112 (2013).
- 6 Ma, P., Yun, J., Deng, H. & Guo, M. Atg1-mediated autophagy suppresses tissue degeneration in pink1/parkin mutants by promoting mitochondrial fission in *Drosophila*. *Molecular biology of the cell* **29**, doi:10.1091/mbc.E18-04-0243 (2018).
- 7 de Castro, I. P. *et al.* *Drosophila* ref(2)P is required for the parkin-mediated suppression of mitochondrial dysfunction in pink1 mutants. *Cell Death Dis* **4**, e873, doi:10.1038/cddis.2013.394 (2013).
- 8 Gehrke, S. *et al.* PINK1 and Parkin control localized translation of respiratory chain component mRNAs on mitochondria outer membrane. *Cell metabolism* **21**, doi:10.1016/j.cmet.2014.12.007 (2015).
- 9 You, Z. *et al.* Requirement for p62 acetylation in the aggregation of ubiquitylated proteins under nutrient stress. *Nature communications* **10**, doi:10.1038/s41467-019-13718-w (2019).
- 10 Nichol, H., Hanson, A. D., Ovsenek, N., Pickering, I. & Juurlink, B. *Iron metabolism in the Drosophila mutants fumble and malvolio* Degree of Master of Science thesis, University of Saskatchewan, (2007).
- 11 Gao, G. & Chang, Y. Z. Mitochondrial ferritin in the regulation of brain iron homeostasis and neurodegenerative diseases. *Frontiers in pharmacology* **5**, doi:10.3389/fphar.2014.00019 (2014).
- 12 Missirlis, F. *et al.* Characterization of mitochondrial ferritin in *Drosophila*. *Proceedings of the National Academy of Sciences of the United States of America* **103**, doi:10.1073/pnas.0601471103 (2006).
- 13 Yang, Y., Wu, Z., Kuo, Y. M. & Zhou, B. Dietary Rescue of Fumble--A *Drosophila* Model for Pantothenate-Kinase-Associated Neurodegeneration. *Journal of inherited metabolic disease* **28**, doi:10.1007/s10545-005-0200-0 (2005).
- 14 Wan, Z. *et al.* Elevating bioavailable iron levels in mitochondria suppresses the defective phenotypes caused by PINK1 loss-of-function in *Drosophila melanogaster*. *Biochemical and biophysical research communications* **532**, doi:10.1016/j.bbrc.2020.08.002 (2020).
- 15 Klopstock, T. *et al.* Fosmetpantotenate Randomized Controlled Trial in Pantothenate Kinase-Associated Neurodegeneration. *Movement disorders : official journal of the Movement Disorder Society* **36**, doi:10.1002/mds.28392 (2021).
- 16 Chang, X. *et al.* Pilot trial on the efficacy and safety of pantethine in children with pantothenate kinase-associated neurodegeneration: a single-arm, open-label study. *Orphanet Journal of Rare Diseases* **15**, 1-9, doi:doi:10.1186/s13023-020-01530-5 (2020).

REVIEWER COMMENTS

Reviewer #1 (Remarks to the Author):

The authors addressed all concerns. I particularly appreciated the authors efforts in providing in depth and accurate investigation of mitophagy.

Reviewer #2 (Remarks to the Author):

In general, the authors did a very good job in addressing many of the comments and concerns and thereby have certainly solidified most aspects of the fly data. However, I am still not entirely convinced by the molecular mechanism as it relates to mitophagy and especially the acetylation of p62 and the resulting enhanced ubiquitin binding. Some of this is only superficially addressed and much remains to be done to further strengthen these specific parts.

Overall, I am very supportive of publication as there are novel and very interesting findings here, but I feel substantially more experimental evidence would be needed for certain claims made. However, I do realize that some of this may be beyond the scope of the current manuscript. Yet at a minimum several aspects need to be more carefully discussed and some statements on mechanisms and relevance to human disease should be toned down. This will help clearly distinguish what is solid and unequivocally demonstrated here from the perhaps more speculative aspects and this should help guide the reader to what remains to be clearly shown. However as is, the manuscript is also already quite dense and somewhat difficult to follow along. While I certainly do appreciate the additional (mitophagy) efforts, I wonder whether a better solution might be to leave out some aspects entirely including (most of) the human cell culture data.

1) Suggested mechanism

For the proposed mechanism as it relates to stronger binding of acetylated Ref(2)P/p62 to ubiquitin, we seem to rely on single immunoblot after immunoprecipitation. There is no quantification of the data, not to mention any real affinity assessment. P62 itself is known to be ubiquitylated and as such it is not even clear whether the presented data is truly reflective of its ubiquitin binding. In my view, this part remains too premature also given that the suggested target lysine residues in the UBA domain residues are not conserved between human and fly p62.

2) Human cells

The human fibroblasts are quite interesting but really should be somewhat expanded to better demonstrate the conservation of effects. This includes protein levels of PINK1, and perhaps Mitofusin1 or 2 levels that may be down regulated. Likewise, PINK2 levels should be found reduced in both PINK1 or Parkin mutant fibroblasts which could really further help translate the fly findings.

However, especially the knockdown approach in HEK cells does not seem to present a clean or robust enough mitophagy model and more thorough analyses would be required here. Neither knockdown of PINK1 nor knockdown or overexpression of PINK2 is clearly discernable from most of the blots shown. Untreated conditions (-CCCp) are not provided. The proteins studied here are all enzymes and as such it will be essential to determine their levels and potentially even study their kinetics over time. Most if not all measures here rely on static colocalization and do not capture the dynamics.

3) Epistasis analysis

While the authors provide additional strong data, it remains somewhat puzzling that parkin mutants do not fully phenocopy all aspects of PINK1 with regards to fbl. I don't think the discussion summarizes these congruent and disparate findings between PINK1 and Parkin well enough. In order to better understand the similarities and distinctions here, a table summarizing the individual and combined genetic and pharmacological manipulations alongside their phenotypic outcomes might be helpful. Perhaps the schematic model can even be further expanded to incorporate the (distinctive) roles of Parkin and its relationship with PINK2.

In that context and especially given the complexity of and difficulties interpreting the epistasis experiments as pointed out by the authors, it will be essential to carefully revise the text to specify complete loss (null allele) versus are reduction of function (knockdown). The terms LOF or deficient seem somewhat interchangeably used but a clear language may help clarify.

4) Other

The last chapter of the discussion appears somewhat out of place. The discussion of statistical testing would be better incorporated elsewhere (materials and methods).

Reviewer #3 (Remarks to the Author):

The revision are satisfactory

Reviewer #4 (Remarks to the Author):

The authors replied appropriately to the criticisms raised by the reviewers and now the data better support the evidence of a novel functional interaction between PINK1 and PANK2. This could lead to a better comprehension of PKAN pathology.

There are a few typos and mistakes that should be addressed:

Page 3 : correct 4-phosphopanthetheine

Page 5: correct lineal pathway

Page 16: correct Fig 6o

Pag 16, last paragraph: I suggest to specify that the cells are exposed to CCCP.

Page 22: correct initiates

Figure 7, panel c and d: I think that there is no overexpression of fbl in PINK1 RNAi flies.

Extended data figure 8, panel b. There are 7 lanes for actin while only 6 lanes for the other target proteins.

Point-by-Point Response (NCOMMS-20-20442-R2)

Reviewer #2 (Remarks to the Author):

In general, the authors did a very good job in addressing many of the comments and concerns and thereby have certainly solidified most aspects of the fly data. However, I am still not entirely convinced by the molecular mechanism as it relates to mitophagy and especially the acetylation of p62 and the resulting enhanced ubiquitin binding. Some of this is only superficially addressed and much remains to be done to further strengthen these specific parts.

We thank Reviewer 2 for his/her appreciation of our efforts. We performed additional experiments to address these points and the results support that acetylation of p62 will enhance ubiquitin binding. Other concerns are also addressed. Please see below for more details.

Overall, I am very supportive of publication as there are novel and very interesting findings here, but I feel substantially more experimental evidence would be needed for certain claims made. However, I do realize that some of this may be beyond the scope of the current manuscript. Yet at a minimum several aspects need to be more carefully discussed and some statements on mechanisms and relevance to human disease should be toned down. This will help clearly distinguish what is solid and unequivocally demonstrated here from the perhaps more speculative aspects and this should help guide the reader to what remains to be clearly shown. However as is, the manuscript is also already quite dense and somewhat difficult to follow along. While I certainly do appreciate the additional (mitophagy) efforts, I wonder whether a better solution might be to leave out some aspects entirely including (most of) the human cell culture data.

We thank Reviewer 2 for these suggestions. We have changed some wordings (in blue) and toned down some relevant statements to describe the disease relevance of our work more objectively.

1) Suggested mechanism

For the proposed mechanism as it relates to stronger binding of acetylated Ref(2)P/p62 to ubiquitin, we seem to rely on single immunoblot after immunoprecipitation. There is no quantification of the data, not to mention any real affinity assessment. P62 itself is known to be ubiquitylated and as such it is not even clear whether the presented data is truly reflective of its ubiquitin binding. In my view, this part remains too premature also given that the suggested target lysine residues in the UBA domain residues are not conserved between human and fly p62.

We toned down our original conclusion by softening our statement. To address the raised concern, in this revision we first performed denaturing IP to check possible direct ubiquitination on the Ref(2)P protein. Interestingly, we failed to detect any positive ubiquitin signal on Ref(2)P proteins in this assay (see below, also Extended Data Fig. 8c), suggesting that the polyubiquitin signal may arise from its ubiquitin binding.

To further assess the binding affinity of acetylated and non-acetylated Ref(2)P proteins to ubiquitin, we conducted *in vitro* acetylation and ubiquitin binding assays using Ref(2)P and TIP60 proteins purified from *E. coli*. We found that acetylated Ref(2)P by TIP60 exhibited significant stronger binding capability to supplied ubiquitin (see below, also Extended Data Figure 8k). These two pieces of data suggest that although the lysine residues in the UBA domain are not conserved between human and fly p62, acetylation on Ref(2)P protein still enhance its binding affinity with ubiquitin during mitophagy.

2) Human cells

The human fibroblasts are quite interesting but really should be somewhat expanded to better demonstrate the conservation of effects. This includes protein levels of PKAN2, Pink1, and perhaps Mitofusin1 or 2 levels that may be down regulated. Likewise, PKAN2 levels should be found reduced in both Pink1 or Parkin mutant fibroblasts which could really further help translate the fly findings.

We attempted to accomplish these experiments suggested by reviewer 2. However, our antibodies gave either multiple bands or low-quality signals on these targets. Thus, we decided not to include them in the manuscript. As for PANK2 expression, we checked its level in *PINK1* mutant cells, but found that the PANK2 protein level was unchanged compared to wild type (see below).

Notwithstanding, we have to point out that similar phenomena were observed before. For example, C-130 expression also remains little altered in *PINK1* mutant fibroblast, but only showed a reduction in converted dopaminergic neurons (Fig.2 of reference paper)¹. One possibility, we speculate, might be that in some cells other gene alteration (such as high levels of Parkin expression) suppressed *PINK1* mutant phenotype. We agree there are issues still unresolved yet in human, and therefore toned down our relevant statements.

However, especially the knockdown approach in HEK cells does not seem to present a clean or robust enough mitophagy model and more thorough analyses would be required here. Neither knockdown of *PINK1* nor knockdown or overexpression of *PKAN* is clearly discernable from most of the blots shown. Untreated conditions (-CCCP) are not provided. The proteins studied here are all enzymes and as such it will be essential to determine their levels and potentially even study their kinetics over time. Most if not all measures here rely on static colocalization and do not capture the dynamics.

To address these issues, in the revised manuscript we provided new *PINK1* and *PANK2* blots to demonstrate the efficiencies of overexpression and knockdown, as well as data from untreated (without CCCP) conditions (see below, also Fig 5h, 5j, 5k and Extended Data Figure 6m, 6n-s, 7b, 7d and 7g-i). As we stated in the METHODS section, after transfection (plasmids of overexpression or knockdown), we treated cells with high dose of antibiotics to enrich for transfected cells based on their selectable markers, so identification of positive cells in the immunofluorescence stain may not be necessary. We also tried to capture the dynamics of mitophagy with mito-Keima indicator, however the imaging facility has been closed for a long time due to the tsunami of COVID-19 cases. While we agree that dynamic analysis will provide more detail and stronger support for our story, we consider our current staining, immunoblotting and genetic evidence already provide strong support for mitophagy involvement.

In (Fig. 5h)

(Fig. 5j)

(Fig. 5l)

(Ext Data Fig. 7b)

(Ext Data Fig. 7d)

3) Epistasis analysis

While the authors provide additional strong data, it remains somewhat puzzling that parkin mutants do not fully phenocopy all aspects of Pink1 with regards to *fbl*. I don't think the discussion summarizes these congruent and disparate findings between Pink1 and Parkin well enough. In order to better understand the similarities and distinctions here, a table summarizing the individual and combined genetic and pharmacological manipulations alongside their phenotypic outcomes might be helpful. Perhaps the schematic model can even be further expanded to incorporate the (distinctive) roles of Parkin and its relationship with PANK2.

In that context and especially given the complexity of and difficulties interpreting the epistasis experiments as pointed out by the authors, it will be essential to carefully revise the text to specify complete loss (null allele) versus are reduction of function (knockdown). The terms LOF or deficient seem somewhat interchangeably used but a clear language may help clarify.

We thank the suggestions. To better understand the consistent and divergent findings between *PINK1* and *parkin*, we provide a table summarizing their interactions with *fbl*, along with further explanations in the DISCUSSION (Table 1). We also more accurately used in the manuscript terminology of "null allele" and "knock down", "Loss-of-function (LOF)" or "deficient" as suggested by the reviewers.

4) Other

The last chapter of the discussion appears somewhat out of place. The discussion of statistical testing would be better incorporated elsewhere (materials and methods).

We include details of the two-proportion Z test in the METHODS section. In the DISCUSSION, we really want to discuss the underlying biological implications behind the choice of this different statistical approach. We moved some sentences to METHODS section to keep our discussion more focused on this issue.

Reviewer #4 (Remarks to the Author):

The authors replied appropriately to the criticisms raised by the reviewers and now the data better support the evidence of a novel functional interaction between PINK1 and PANK2. This could lead to a better comprehension of PKAN pathology. We appreciate the reviewers' acknowledgment of our efforts. We have corrected the following typos in the revised manuscript, except for Extended Data Figure 6o on page 16 where we are not sure where the error is.

There are a few typos and mistakes that should be addressed:

Page 3: correct 4-phosphopanthetheine

Page 5: correct lineal pathway

Page 16: correct Fig 6o

Pag 16, last paragraph: I suggest to specify that the cells are exposed to CCCP.

Page 22: correct initiates

Figure 7, panel c and d: I think that there is no overexpression of fbl in PINK1 RNAi flies.

Extended data figure 8, panel b. There are 7 lanes for actin while only 6 lanes for the other target proteins.

Reference:

1. Gehrke, S. *et al.* PINK1 and Parkin control localized translation of respiratory chain component mRNAs on mitochondria outer membrane. *Cell metabolism* **21** (2015).

REVIEWERS' COMMENTS

Reviewer #2 (Remarks to the Author):

The authors addressed my remaining concerns. I appreciate their additional efforts and do recommend publication of this very interesting study in Nature Communications.